# HALP: HARDWARE-AWARE LATENCY PRUNING

## ABSTRACT

Structural pruning can simplify network architecture and improve inference speed. We propose Hardware-Aware Latency Pruning (HALP) that formulates structural pruning as a global resource allocation optimization problem, aiming at maximizing the accuracy while constraining latency under a predefined budget. For filter importance ranking, HALP leverages latency lookup table to track latency reduction potential and global saliency score to gauge accuracy drop. Both metrics can be evaluated very efficiently during pruning, allowing us to reformulate global structural pruning under a reward maximization problem given target constraint. This makes the problem solvable via our augmented knapsack solver, enabling HALP to surpass prior work in pruning efficacy and accuracy-efficiency trade-off. We examine HALP on both classification and detection tasks, over varying networks, on ImageNet and VOC datasets. In particular, for ResNet-50/-101 pruning on ImageNet, HALP improves network throughput by $1.60\times/1.90\times$ with $+0.3\%/-0.2\%$ top-1 accuracy changes, respectively. For SSD pruning on VOC, HALP improves throughput by $1.94\times$ with only a 0.56 mAP drop. HALP consistently outperforms prior art, sometimes by large margins.

## 1 INTRODUCTION

Convolutional Neural Networks (CNNs) act as the central tenet behind the rapid development in computer vision tasks such as classification, detection, segmentation, image synthesis, among others. As performance boosts, so do model size, computation, and latency. With millions, sometimes billions of parameters (*e.g.*, GPT-3 Brown et al. (2020)), modern neural networks face increasing challenges upon ubiquitous deployment, that mostly faces stringent constraints such as energy and latency Dai et al. (2019); Molchanov et al. (2016; 2019). In certain cases like autonomous driving, a breach of real-time constraint not only undermines user experience, but also imposes critical safety concerns. Even for cloud service, speeding up the inference of neural networks directly translates into higher throughput, allowing more clients and users to benefit from the service.

One effective and efficient method to reduce model complexity is through network pruning. The primary goal of pruning is to remove the parameters, along with their computation, that are deemed least important for inference Alvarez & Salzmann (2016); Han et al. (2015); Liu et al. (2019b); Molchanov et al. (2019). Compatible with other compression streams of work such as quantization Cai et al. (2020); Wang et al. (2020b); Zhu et al. (2016) and distillation Hinton et al. (2015); Mullapudi et al. (2019); Yin et al. (2020), network pruning enables a flexible tuning of model complexity towards varying constraints, while requiring much less design efforts by neural architecture search Tan et al. (2019); Vahdat et al. (2020); Wu et al. (2019) and architecture re-designs Howard et al. (2017); Ma et al. (2018); Tan & Le (2019). Thus, in this work, we study pruning, in particular structured pruning that removes filters (or neurons) to benefit off-the-shelf platforms, *e.g.*, GPUs.

As the pruning literature develops, the pruning criteria also evolves to better reflect final efficiency. The early phase of the field focuses on maximum parameter removal in seek for minimum representations of the pretrained model. This leads to a flourish of approaches that rank neurons effectively to measure their importance Molchanov et al. (2016); Wang et al. (2020a). As each neuron/filter possesses intrinsically different computation, following works explore proxy to enhance redundancy removal, FLOPs being one of the most widely adopted metrics Li et al. (2020); Wu et al. (2020); Yu & Huang (2019) to reflect how many multiplication and addition computes needed for the model. Hovever, for models with very similar FLOPs, their latency can vary significantly Tan et al. (2019). Recently, more and more works start directly working on reducing latency Chen et al. (2018); Yang

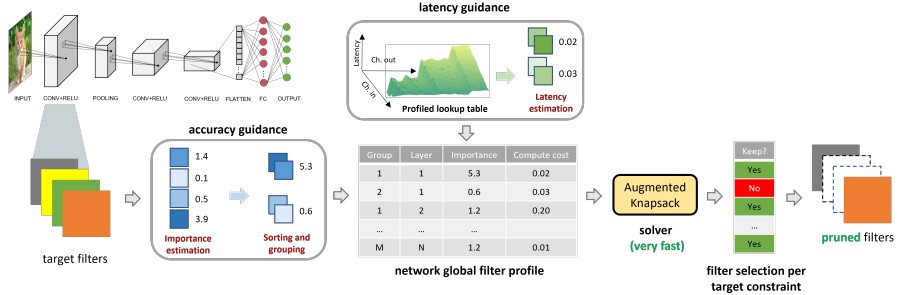

Figure 1: The proposed hardware-aware latency pruning (HALP) paradigm. Considering both performance and latency contributions, HALP formulates global structural pruning as a global resource allocation problem (Section 3.1), solvable using our augmented Knapsack algorithm (Section 3.2). Pruned architectures surpass prior work across varying latency constraints given changing network architectures for both classification and detection tasks (Section 4).

et al. (2018). However, not much was done in the field of GPU friendly pruning methods due to non-trivial latency-architecture trade-off. For example, as recently observed in Radu et al. (2019), GPU usually imposes staircase-shaped latency patterns for convolutional operators with varying channels, which inevitably occur per varying pruning rate, see the latency surface in Fig. 1. This imposes a constraint that pruning needs to be done in groups to achieve latency improvement. Moreover, getting the exact look-up table of layers under different pruning configuration will benefit maximizing performance while minimizing latency.

Pruning different layers in the deep neural network will result in different accuracy-latency trade-off. Typically, removing channels from the latter layers has smaller impact on accuracy and smaller impact on latency versus removing channels from the early layers. We ask the question, if it is better to remove *more* neurons from latter layer or *less* from early layer to achieve the same accuracy-latency trade-off. By nature, the problem is combinatorial and requires the appropriate solution.

In this paper, we propose hardware-aware latency pruning (HALP) that formulates pruning as a resource allocation optimization problem to maximize the accuracy while maintaining a latency budget. The overall workflow is shown in Fig. 1. For latency estimate per pruned ar-

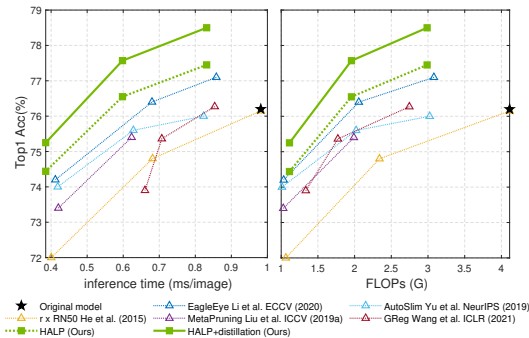

Figure 2: Pruning ResNet50 on the ImageNet dataset. The proposed HALP surpasses state-of-the-art structured pruning methods over accuracy, latency, and FLOPs metrics. Target hardware is NVIDIA Titan V GPU. **Top-left** is better.

chitecture, we pre-analyze the operator-level latency values by creating a look-up table for every layer of the model on the target hardware. Then we introduce an additional score for each neuron group to reflect and encourage latency reduction. To this end, we first rank the neurons according to their importance estimates, and then dynamically adjust their latency contributions. With neurons re-calibrated towards the hardware-aware latency curve, we now select remaining neurons to maximize the gradient-based importance estimates for accuracy, within the total latency constraint. This makes the entire neuron ranking solvable under the knapsack paradigm. To enforce the neuron selection order in a layer to be from the most important to the least, we have enhanced the knapsack solver so that the calculated latency contributions of the remaining neurons would hold. HALP surpasses prior art in pruning efficacy, see Fig. 2 and the more detailed analysis in Section 4.

Our main contributions are summarized as follows:

- We propose a latency-driven structured pruning algorithm that exploits hardware latency traits to yield direct inference speedups.
- We orient the pruning process around a quick yet highly effective knapsack scheme that seeks for a combination of remaining neuron groups to maximize importance while constraining to the target latency.

- We introduce a group size adjustment scheme for knapsack solver amid varying latency contributions across layers, hence allowing full exploitation of the latency landscape of the underlying hardware.
- We compare to prior art when pruning ResNet, MobileNet, VGG architectures on ImageNet, PASCAL VOC and demonstrate that our method yields consistent latency and accuracy improvements over state-of-the-art methods. Our ImageNet pruning results present a viable $1.6\times$ to $1.9\times$ speedup while preserving very similar original accuracy of the ResNets.

## 2 RELATED WORK

**Pruning methods**. Depending on when to perform pruning, current methods can generally be divided into three groups Frankle & Carbin (2019b): i) prune pretrained models Han et al. (2015); Luo et al. (2017); Li et al. (2017); He et al. (2018); Molchanov et al. (2016; 2019); Gale et al. (2019), ii) prune at initialization Frankle & Carbin (2019a); Lee et al. (2019); de Jorge et al. (2020), and iii) prune during training Alvarez & Salzmann (2016); Gao et al. (2019); Lym et al. (2019). Despite notable progresses in the later two streams, pruning pretrained models remains as the most popular paradigm, with structural sparsity favored by off-the-shelf inference platforms such as GPU.

To improve on inference efficiency, many previous pruning methods trim down the neural network aiming to achieve a high compression rate while maintaining an acceptable accuracy. The estimation of neuron importance has been widely studied in literature Hu et al. (2016); Luo et al. (2017); Molchanov et al. (2016). For example, Molchanov et al. (2019) proposes to use Taylor expansion to measure the importance of neurons and prunes the least-ranked ones until a desired number of neurons are pruned. However, a compression ratio does not directly translate into computation reduction ratio, amid the fact that each neuron/filter possesses different computation.

There are recent methods that focus primarily on reducing FLOPs. Some of them take FLOPs into consideration when calculating the neuron importance to encourage penalizing neurons that induce high computations Wu et al. (2020). An alternative line of work propose to select the best pruned network from a set of candidates Li et al. (2020); Yang et al. (2018). However, it would take a long time for candidate selection due to the large amount of candidates. In addition, these methods use FLOPs as a proxy of latency, which is usually inaccurate as networks with similar FLOPs might have significantly different latencies Tan et al. (2019).

**Latency-aware compression.** Emerging compression techniques shift attention to directly prune to cut down on latency. One popular stream is Neural Architecture Search (NAS) methods Dai et al. (2019); Dong et al. (2018); Tan et al. (2019); Wu et al. (2019) that adaptively adjusts the architecture of the network for a given latency requirement. They incorporate the platform constraints into the optimization process in both the architecture and parameter space to jointly optimize the model size and accuracy. Despite remarkable insights, NAS methods remain computationally expensive in general compared to their pruning counterparts.

Latency-oriented pruning has also gained a growing amount of attention. Chen et al. (2018) presents a framework for network compression under operational constraints, using Bayesian optimization to iteratively obtain compression hyperparameters that satisfies the constraints. Along the same line, NetAdapt Yang et al. (2018) iteratively prunes neurons across layers under the guidance of the empirical latency measurements on the targeting platform. While these methods push the frontier of latency constrained pruning, the hardware-incurred latency surface in fact offers much more potential under our enhanced pruning policy - as we show later, large rooms for improvements remain un-exploited and realizable.

## 3 METHOD

In this section, we first formulate the pruning process as an optimization process, before diving deep into the importance estimation for accuracy and latency. Then, we elaborate on how to solve the optimization via knapsack regime, augmented by dynamic grouping of neurons. We finalize the method by combining these key steps under one realm of HALP.

### 3.1 Objective Function

Consider a neural network that consists of $L$ layers performing linear operations on their inputs, together with non-linear activation layers and potentially pooling layers. Suppose there are $N_l$ neurons (output channels) in the $l_{\text{th}}$ layer and each neuron is encoded by parameters $\mathbf{W}_l^n \in \mathbb{R}^{C_l^{in} \times K_l \times K_l}$, where $C^{in}$ is the number of input channels and $K$ is the kernel size. By putting all the neurons across the network together, we get the neuron parameter set $\mathbf{W} = \{\{\mathbf{W}_l^n\}_{n=1}^{N_l}\}_{l=1}^{L}$, where $N = \sum_{l=1}^{L} N_l$ is the total number of neurons in the network.

Given a training set $\mathcal{D} = \{(x_i, y_i)\}_{i=1}^{M}$, the problem of network pruning with a given constraint $C$ can be generally formulated as the following optimization problem:

$$\arg\min_{\hat{\mathbf{W}}} \mathcal{L}(\hat{\mathbf{W}}, \mathcal{D}) \quad \text{s.t.} \quad \Phi\left(f(\hat{\mathbf{W}}, x_i)\right) \leq C \tag{1}$$

where $\hat{\mathbf{W}} \subset \mathbf{W}$ is the set of remaining neurons after pruning and $\mathcal{L}$ is the loss of the task. $f(\cdot)$ encodes the network function, and $\Phi(\cdot)$ maps the network to the constraint $C$, such as latency, FLOPs, or memory. We primarily focus on latency for $\Phi(\cdot)$ in this work while the method easily scales to other constraints.

The key to solving the aforementioned problem relies on identifying the portion of the network that satisfies the constraint while incurring minimum performance disruption:

$$\arg\max_{p_1,\cdots,p_l} \sum_{l=1}^{L} I_l(p_l), \quad \text{s.t.} \quad \sum_{l=1}^{L} T_l(p_{l-1}, p_l) \leq C, \ \forall l \ \ 0 \leq p_l \leq N_l, \tag{2}$$

where $p_l$ denotes the number of kept neurons at layer $l$, $I_l(p_l)$ signals the maximum importance to the final accuracy with $p_l$ neurons, and $T_l(p_{l-1}, p_l)$ checks on the associated latency contribution of layer $l$ with $p_{l-1}$ input channels and $p_l$ output channels. $p_0$ denotes a fixed input channel number for the first convolutional block, *e.g.*, 3 for RGB images. We next elaborate on $I(\cdot)$ and $T(\cdot)$ in detail.

**Importance score.** To get the importance score of a layer to final accuracy, namely $I_l(p_l)$ in Eq. 2, we take it as the accumulated score from individual neurons $\sum_{j=1}^{p_l} \mathcal{I}_l^j$. We first approximate the importance of neurons using the Taylor expansion of the loss change Molchanov et al. (2019). Specifically, we prune on batch normalization layers and the importance of the $n$-th neuron in the $l$-th layer is calculated as

$$\mathcal{I}_l^n = \left| g_{\gamma_l^n} \gamma_l^n + g_{\beta_l^n} \beta_l^n \right|, \tag{3}$$

where $g$ denotes the gradient of the weight, $\gamma_l^n$ and $\beta_l^n$ are the corresponding weight and bias from the batch normalization layer, respectively. Unlike a squared loss in Molchanov et al. (2019), we use absolute difference as we observe slight improvements.

In order to maximize the total importance, we keep the most important neurons at a higher priority. To this end, we rank the neurons in the $l_{th}$ layer according to their importance score in a descending order and denote the importance score of the $j_{th}$-ranked neuron as $\mathcal{I}_l^j$, thus we have

$$I_l(p_l) = \sum_{j=1}^{p_l} \mathcal{I}_l^j, \quad 0 \leq p_l \leq N_l, \ \mathcal{I}_l^1 \geq \cdots \geq \mathcal{I}_l^{N_l}. \tag{4}$$

**Latency contribution.** We empirically obtain the layer latency $T_l(p_{l-1}, p_l)$ in Eq. 2 by pre-building a layer-wise look-up table with pre-measured latencies. This layer latency corresponds to the aggregation of the neuron latency contribution of each neuron in the layer, $c_l^j$:

$$T_l(p_{l-1}, p_l) = \sum_{j=1}^{p_l} c_l^j, \quad 0 \leq p_l \leq N_l. \tag{5}$$

The latency contribution of the $j$-th neuron in the $l$-th layer can also be computed using the entries in the look up table as:

$$c_l^j = T_l(p_{l-1}, j) - T_l(p_{l-1}, j-1), \quad 1 \leq j \leq p_l. \tag{6}$$

---

**Algorithm 1** Augmented Knapsack Solver

**Input:** Importance score $\{\mathcal{I}_l \in \mathbb{R}^{N_l}\}_{l=1}^L$ where $\mathcal{I}_l$ is sorted descendingly; Neuron latency contribution $\{c_l \in \mathbb{R}^{N_l}\}_{l=1}^L$; Latency constraint $C$.

1: $\text{maxV} \in \mathbb{R}^{(C+1)}$, $\text{keep} \in \mathbb{R}^{L \times (C+1)}$      ▷ maxV[c]: max importance under constraint $c$; keep[l, c]: # neurons to keep in layer $l$ to achieve maxI[c]
2: **for** $l = 1, \ldots, L$ **do**
3:      **for** $j = 1, \ldots, N_l$ **do**
4:          **for** $c = 1, \ldots, C$ **do**
5:              $v_{keep} = \mathcal{I}_l^j + \text{maxV}[c - c_l^j]$, $v_{prune} = \text{maxV}[c]$    ▷ total importance can achieve under constraint $c$ with object $n$ being kept or not
6:              **if** $v_{keep} > v_{prune}$ **and** $\text{keep}[l, c - c_l^j] == j - 1$ **then**    ▷ check if it leads to higher score and if more important neurons in layer are kept
7:                  $\text{keep}[l, c] = j$, $\text{update\_maxV}[c] = v_{keep}$
8:              **else**
9:                  $\text{keep}[l, c] = \text{keep}[l, c - 1]$, $\text{update\_maxV}[c] = v_{prune}$
10:              **end if**
11:          **end for**
12:          $\text{maxV} \leftarrow \text{update\_maxV}$
13:      **end for**
14: **end for**
15:
16: $\text{keep\_n} = \varnothing$ to save the kept neurons in model
17: **for** $l = L, \ldots, 1$ **do**              ▷ retrieve the set of kept neurons
18:      $p_l = \text{keep}[l, C]$
19:      $\text{keep\_n} \leftarrow \text{keep\_n} \cup \{p_l \text{ top ranked neurons in layer } l\}$
20:      $C \leftarrow C - \sum_{j=1}^{p_l} c_l^j$
21: **end for**

**Output:** Kept important neurons (keep\_n).

---

In practice, we first rank globally neurons by importance and then consider their latency contribution. Thus, we can draw the following properties. If we remove the least important neuron in layer $l$, then the number of neurons will change from $p_l$ to $p_l - 1$, leading to a latency reduction $c_l^{p_l}$ as this neuron's latency contribution score. We assign the potential latency reduction to neurons in the layer by the importance order. The most important neuron in that layer would always have a latency contribution $c_l^1$. At this stage, finding the right combination of neurons to keep imposes a combinatorial problem, and in the next section we tackle it via reward maximization considering latency and accuracy traits.

### 3.2 AUGMENTED KNAPSACK SOLVER

Given both importance and latency estimates, we now aim at solving Eq. 2. By plugging back in the layer importance Eq. 4 and layer latency Eq. 5, we come to

$$\max \sum_{l=1}^L \sum_{j=1}^{p_l} \mathcal{I}_l^j, \quad \text{s.t.} \quad \sum_{l=1}^L \sum_{j=1}^{p_l} c_l^j \leq C, \quad 0 \leq p_l \leq N_l, \ \mathcal{I}_l^1 \geq \mathcal{I}_l^2 \geq \ldots \mathcal{I}_l^{N_l}. \tag{7}$$

This simplifies the overall pruning process into a knapsack problem only with additional preceding constraints. The preceding enforcement originates from the fact that for a neuron with rank $j$ in the $l_{th}$ layer, the neuron latency contribution only holds when all the neurons with rank $r = 1, \ldots, j - 1$ are kept in the $l_{th}$ layer and the rest of the neurons with rank $r = j + 1, j + 2, \cdots, p_l$ are removed. Yet the problem is solvable by specifying each item with a list of preceding items that need to be selected before its inclusion.

We augment the knapsack solver to consider the reordered neurons with descending importance score so that all the preceding neurons will removed before it. A detailed description of the pseudo code of the augmented knapsack solver is provided in Algo. 1. The augmented solver is required to make sure that the latency cost is correct.

### 3.3 NEURON GROUPING

Considering each neuron individually results in burdensome computation during pruning. We next explore grouping neurons so that a number of them can be jointly considered and removed enabling faster pruning Yin et al. (2020). This helps exploit hardware-incurred channel granularity guided by the latency, and speed up knapsack solving of Eq. 7.

We refer to the difference of neurons counts between two latency cliffs of the staircase-patterned latency as the latency step size. In our method, we group $s$ channels in a layer as an entirety, where the value of $s$ is equal to the latency step size. The neurons are grouped by the order of importance. Then we aggregate the importance score and latency contribution for the grouped entity. When dealing with skip connections in ResNet and group convolutions in MobielNet, we not only group

neurons within a layer, we also group the neurons sharing the same channel index from the connected layers Ding et al. (2019); Luo & Wu (2020). Note that the latency step size for different layers might be different. For cross-layer grouping, we use the largest group size among the layers. Latency-aware grouping enables additional performance benefits when compared to a heuristic universal grouping, as we will later show in the experiments.

One noticeable benefit of neuron grouping is the simplification of knapsack problem that scales linearly with the number of candidates under consideration (see Line 2 of Alg. 1). As an example, by grouping neurons together for a ResNet50, the total number of (grouped) neurons can be greatly reduced from $26, 560$ to only $215$, this drastically speedups the solver. Such saving can be especially beneficial as network size scales up, a very likely trend that literature foresees.

### 3.4 FINAL HALP REGIME

With all aforementioned steps, we formulate the final HALP as follows. The pruning process takes a trained network as input and prunes it iteratively to satisfy the requirement of a given latency budget $C$. We perform one pruning every $r$ minibatches and repeat it $k$ pruning steps in total. In particular, we set $k$ milestones gradually decreasing the total latency to reach the goal via exponential scheduler de Jorge et al. (2020), with $C^1 > C^2 > \cdots > C^k$, $C^k = C$. The algorithm gradually trims down neurons using steps below:

- **Step 1**. For each minibatch, we get the gradients of the weights and update the weights as during the normal training. We also calculate each neuron's importance score as Eq. 3.
- **Step 2**. Over multiple minibatches we calculate the average importance score for each neuron and rank them accordingly. Then we count the number of neurons remaining in each layer and dynamically adjust the latency contribution as in Eq. 6.
- **Step 3**. We group neurons as described in Sec. 3.3 and calculate group's importance and latency reduction. Then we get the nearest preceding group for each layer.
- **Step 4**. We execute the Algo. 1 to select the neurons being remained with current latency milestone. Repeat starting from the Step 1 until $k$ milestones are reached.

Once pruning finishes we fine-tune the network to recover accuracy.

## 4 EXPERIMENTS

We demonstrate the efficacy and feasibility of the proposed HALP method in this section. We use ImageNet ILSVRC2021 Russakovsky et al. (2015) for classification. We first study the impact of grouping size $s$ on the pruned top-1 accuracy and the inference time to show the effectiveness of our grouping scheme. We then study four architectures (ResNet50, ResNet101, VGG16 and MobileNet-V1/V2) on different GPUs and compare our pruning results with the state-of-the-art methods on classification task. Finally, we further show the generalization ability of our algorithm by testing with object detection task. We introduce the details of experimental setting in appendix Sec. A. Unless otherwise stated, we target a NVIDIA TITAN V GPU for main experiments and measure latency at batch size 256 for ResNets and VGG, and 512 for MobileNets. We include experimental results for smaller batch size in the appendix, which also show the efficacy of our algorithm.

### 4.1 RESULTS ON IMAGENET

To demonstrate the effectiveness of the proposed HALP method, we compare HALP with several state-of-the-art methods on the large scale dataset ImageNet.

**ResNets.** We start by pruning ResNet50 and ResNet101 and compare our results with state-of-the-art methods in Tab. 1 on TITAN V. In order to have a fair comparison of the latency, for all the other methods, we recreate pruned networks according to the pruned structures they published and measure the latency. Those methods showing '-' in the table do not have pruned structures published so we are unable to measure the latency. For our method, by setting the percentage of latency to remain after pruning to be $X$, we get the final pruned model and refer to it as HALP-$X$%. We report FPS (frames per second) in the table and calculate the speedup of a pruned network as the ratio of FPS between pruned and unpruned models.

| Method | FLOPs (G) | Top1 (%) | Top5 (%) | FPS (im/s) | Speedup |
|---|---|---|---|---|---|
| **ResNet50** | | | | | |
| No pruning | 4.1 | 76.2 | 92.87 | 1019 | 1× |
| ThiNet-70 Luo et al. (2017) | 2.9 | 75.8 | 90.67 | - | - |
| AutoSlim Yu & Huang (2019) | 3.0 | 76.0 | - | 1215 | 1.14× |
| MetaPruning Liu et al. (2019a) | 3.0 | 76.2 | - | - | - |
| GReg-1 Wang et al. (2021) | 2.7 | 76.3 | - | 1171 | 1.15× |
| **HALP**-80% **(Ours)** | 3.1 | **77.2** | **93.47** | 1256 | 1.23× |
| 0.75× ResNet50 He et al. (2015) | 2.3 | 74.8 | - | 1467 | 1.44× |
| ThiNet-50 Luo et al. (2017) | 2.1 | 74.7 | 90.02 | - | - |
| AutoSlim Yu & Huang (2019) | 2.0 | 75.6 | - | 1592 | 1.56× |
| MetaPruning Liu et al. (2019a) | 2.0 | 75.4 | - | 1604 | 1.58× |
| GBN You et al. (2019) | 2.4 | 76.2 | 92.83 | - | - |
| CAIE Wu et al. (2020) | 2.2 | 75.6 | - | - | - |
| LEGR Chin et al. (2020) | 2.4 | 75.6 | 92.70 | - | - |
| GReg-2 Wang et al. (2021) | 1.8 | 75.4 | - | 1414 | 1.39× |
| **HALP**-55% **(Ours)** | 2.0 | **76.5** | **93.05** | 1630 | 1.60× |
| 0.50× ResNet50 He et al. (2015) | 1.1 | 72.0 | - | 2498 | 2.45× |
| ThiNet-30 Luo et al. (2017) | 1.2 | 72.1 | 88.30 | - | - |
| AutoSlim Yu & Huang (2019) | 1.0 | 74.0 | - | 2390 | 2.45× |
| MetaPruning Liu et al. (2019a) | 1.0 | 73.4 | - | 2381 | 2.34× |
| CAIE Wu et al. (2020) | 1.3 | 73.9 | - | - | - |
| GReg-2 Wang et al. (2021) | 1.3 | 73.9 | - | 1514 | 1.49× |
| **HALP**-30% **(Ours)** | 1.0 | **74.3** | **91.81** | 2755 | 2.70× |
| **ResNet50 - EagleEye Li et al. (2020) baseline** | | | | | |
| No pruning | 4.1 | 77.2 | 93.70 | 1019 | 1× |
| EagleEye-3G Li et al. (2020) | 3.0 | 77.1 | 93.37 | 1165 | 1.14× |
| **HALP**-80% **(Ours)** | 3.0 | **77.5** | **93.60** | 1203 | 1.18× |
| EagleEye-2G Li et al. (2020) | 2.1 | 76.4 | 92.89 | 1471 | 1.44× |
| **HALP**-55% **(Ours)** | 2.1 | **76.6** | **93.16** | 1672 | 1.64× |
| EagleEye-1G Li et al. (2020) | 1.0 | 74.2 | 91.77 | 2429 | 2.38× |
| **HALP**-30% **(Ours)** | 1.2 | **74.5** | **91.87** | 2597 | 2.55× |

| Method | FLOPs (G) | Top1 (%) | FPS (im/s) | Speedup |
|---|---|---|---|---|
| **ResNet101** | | | | |
| No pruning | 7.8 | 77.4 | 620 | 1× |
| Taylor-75% Molchanov et al. (2019) | 4.7 | 77.4 | 750 | 1.21× |
| **HALP**-60% **(Ours)** | 4.3 | **78.3** | 847 | 1.37× |
| **HALP**-50% **(Ours)** | 3.6 | 77.8 | 994 | 1.60× |
| Taylor-55% Molchanov et al. (2019) | 2.9 | 76.0 | 908 | 1.47× |
| **HALP**-40% **(Ours)** | 2.7 | **77.2** | 1180 | 1.90× |
| **HALP**-30% **(Ours)** | 2.0 | 76.5 | 1521 | 2.45× |
| **VGG-16** | | | | |
| No pruning | 15.5 | 71.6 | 766 | 1× |
| FBS-3× Gao et al. (2018) | 5.1 | 71.2 | - | - |
| **HALP**-30% **(Ours)** | 4.6 | **72.3** | 1498 | 2.42× |
| FBS-5× Gao et al. (2018) | 3.0 | 70.5 | - | - |
| **HALP**-20% **(Ours)** | 2.8 | **70.8** | 1958 | 5.49× |

| Method | FLOPs (M) | Top1 (%) | FPS (im/s) | Speedup |
|---|---|---|---|---|
| **MobileNet-V1** | | | | |
| No pruning | 569 | 72.6 | 3415 | 1× |
| MetaPruning Liu et al. (2019a) | 142 | 66.1 | 7050 | 2.06× |
| AutoSlim Yu & Huang (2019) | 150 | 67.9 | 7743 | 2.27× |
| **HALP**-42% **(Ours)** | 171 | **68.3** | 7940 | 2.32× |
| 0.75× MobileNetV1 | 325 | 68.4 | 4678 | 1.37× |
| AMC He et al. (2018) | 285 | 70.5 | 4857 | 1.42× |
| NetAdapt Yang et al. (2018) | 284 | 69.1 | - | - |
| MetaPruning Liu et al. (2019a) | 316 | 70.9 | 4838 | 1.42× |
| EagleEye Li et al. (2020) | 284 | 70.9 | 5020 | 1.47× |
| **HALP**-60% **(Ours)** | 297 | **71.3** | 5754 | 1.68× |
| **MobileNet-V2** | | | | |
| No pruning | 301 | 72.1 | 3080 | 1× |
| **HALP**-60% **(Ours)** | 183 | 70.4 | 5668 | 1.84× |
| **HALP**-75% **(Ours)** | 249 | 72.2 | 4110 | 1.33× |

Table 1: ImageNet structural pruning results. We compare HALP for ResNet50 with two different dense baselines (left), ResNet101 and VGG16 (right up), MobileNet-V1 and MobileNet-V2 (right bottom) pruning experiments, with detailed comparison to state-of-the-art pruning methods over varying performance metrics.

From the results comparison we can find that for pruned networks with similar FLOPs using different methods, our method achieves the highest accuracy and also the fastest inference speed. This also shows that FLOPs do not correspond 1:1 to the latency. Among these methods for ResNet50 comparison, EagleEye Li et al. (2020) yields the closest accuracy to ours, but the speedup is lower than ours. In Table. 1, for the pruned ResNet50 network with 3G FLOPs remaining, our method achieves a .4% higher top1 accuracy and slightly (.04×) faster inference. It is expected that the advantage of our method for accelerating the inference is more obvious when it comes to a more compact pruned network, which is 14% (or .20×) additionally faster for a 2G-FLOPs network while increasing accuracy by .2%. When we further prune the network to 1G, we get a pruned network that has more FLOPs but still faster inference speed (.17×) compared to EagleEye-1G, while we also get .3% higher accuracy. We analyze the pruned network structure in detail in the supplementary material. We plot the results comparison in Fig. 2, where we also add the results of training our pruned network with a teacher model RegNetY-16GF (top1 82.9%) Radosavovic et al. (2020). With knowledge distillation, our model is 2.70× faster than the original model at 1% accuracy drop.

**Scalability to other networks.** We next experiment with other models including both VGG Simonyan & Zisserman (2015), MobileNetV1 Howard et al. (2017) and MobileNetV2 Sandler et al. (2018). Same as pruning on ResNets, we perform pruning with many different latency constraints and compare with prior art in Tab. 1. As shown, among these methods, the proposed HALP performs significantly better with higher top-1 accuracy and larger inference speedup.

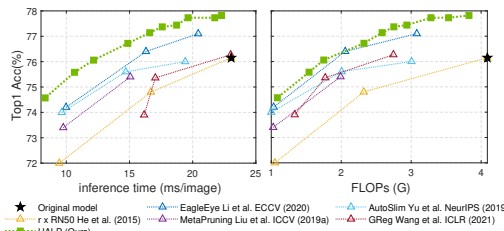

Figure 3: Pruning ResNet50 on the ImageNet dataset with Jetson TX2. Top-left is better.

**Scalability to other hardware.** Our approach is not limited to a single platform. We can compute the latency look-up table to apply HALP to a different platform. We repeat the ResNet50 experiments as described earlier on a Jetson TX2 and compare our results with other methods. The latency is measured with a batch size 32. As shown in Fig.3, in a Jetson TX2, our approach presents faster inference time while maintaining similar FLOPs and better accuracy.

## 4.2 HALP Acceleration on GPUs with TensorRT

To make it closer to the real application in production, we also export the models into onnx format and test the inference speed with TensorRT. We run the inference of the model with FP32, FP16 and also INT8. For INT8, we quantize the model using entropy calibration with 2560 randomly selected ImageNet training images. Since the INT8 TensorCore speedup is not supported in TITAN V GPU, we only report the quantized results on RTX3080 GPU. The accelerations and the corresponding top1 accuracy drop (compared to PyTorch baseline model) are listed in Tab. 2. We can further build a TensorRT INT8 lookup table to achieve potentially larger speedup, which remains as our future work.

| model | acc drop | TITAN V GPU | | RTX3080 GPU | | |
|---|---|---|---|---|---|---|
| | | FP32 | FP16 | FP32 | FP16 | INT8 (acc drop) |
| EagleEye-3G | −0.90% | 1.14× | 4.26× | 1.06× | 3.09× | 6.31× (−0.84%) |
| **HALP-**80% **(Ours)** | −1.25% | **1.24×** | **4.70×** | **1.18×** | **3.32×** | **6.40×** (−1.02%) |
| EagleEye-2G | −0.18% | 1.54× | 5.10× | 1.35× | 3.68× | 7.46× (0.27%) |
| **HALP-**55% **(Ours)** | −0.35% | **1.80×** | **6.36×** | **1.68×** | **4.45×** | **9.14×** (0.13%) |
| EagleEye-1G | 2.02% | 2.73× | 7.81× | 2.29× | 5.61× | 12.29× (2.55%) |
| **HALP-**30% **(Ours)** | 1.76% | **2.91×** | **9.61×** | **2.56×** | **6.44×** | **14.12×** (2.38%) |

Table 2: HALP acceleration of ResNet50 on GPUs with TensorRT (version 7.2.1.6).

## 4.3 Design Effort for Pruning

In addition to noticeable performance boosts, HALP in fact requires less design effort compared to prior art, as summarized in Tab. 3 (details in Appendix O). NetAdapt Yang et al. (2018) and AutoSlim Yu & Huang (2019) generate many proposals during iterative pruning. Then evaluations of the proposals are needed to select the best candidate that will proceed to the next pruning iteration. EagleEye Li et al. (2020) pre-obtains 1000 candidates before pruning and evaluates all of them in order to get the best one. Such pruning candidate selection is intuitive but causes a lot of additional time costs. The computation cost for MetaPruning Liu et al. (2019a) and AMC He et al. (2018) can be even higher because they need to train auxiliary network to generate the pruned structure.

Compared to these methods, our method does not require auxiliary network training nor sub-network evaluation. The latency contribution in our method can be quickly obtained during pruning by the pre-generated latency lookup table. Although creating the table for the target platform might cost time, we only do it once for all pruning ratios. Solving the augmented knapsack problem brings extra computation, however, after neuron grouping, it only takes around additional 30 minutes of CPU time in total for ResNet50 pruning and less than 1 minute for

| Method | Evaluate proposals? | Auxiliary net training? | Sub-network selection (RseNet50) |
|---|---|---|---|
| NetAdapt | Y | N | ∼ 195h (GPU) |
| ThiNet | Y | N | ∼ 210h (GPU) |
| EagleEye | Y | N | 30h (GPU) |
| AutoSlim | Y | Y | |
| MetaPruning | Y | Y | |
| AMC | N | Y | |
| **HALP (Ours)** | **N** | **N** | 6.5**h (GPU)** + 0.5**h (CPU)** |

Table 3: Comparison of extra computation required by pruning methods on ImageNet. Our approach is around *4.3× faster* than the next best method. Sub-network selection timing is approximated as running on same device (a NVIDIA V100).

MobileNetV1, which is negligible compared to the fine-tuning process or training additional models. Moreover, this is significantly lower than other methods, for example the fastest of them EagleEye Li et al. (2020) requires 30 GPU hours.

## 4.4 Efficacy of Neuron Grouping

We first show the benefits of latency-aware neuron grouping and compare the performance under different group size settings.

**Performance comparison.** As described in Sec. 3.3, we group $s$ neurons in a layer as an entirety so that they are removed or kept together. Choosing different group sizes can lead to different performances, and also different computation cost on the augmented knapsack problem solving. In our method, we set an individual group size for each layer according to each layer's latency step size in the look-up table. We name the grouping in our method as latency-aware grouping (LG). For instance, for a ResNet50, using this approach we set the individual group size of 23 layers to 32, of 20 layers to 64, and 10 layers to 128. Layers closer to the input tend to use a smaller group size. Another option for neuron grouping is to heuristically set a fixed group size for all layers as literature does Yin et al. (2020).

Fig. 4 shows the performance of our grouping approach compared to various fixed group sizes for a ResNet50 pruned with different constraints. As shown, using small group sizes yields the worst performance independently of the latency constraint. At the same time, a very large group such as

256 do also harms the final performance. Intuitively, a large group size averages the contribution of many neurons and therefore is not discriminative enough to select the most important ones. Besides, large groups might promote pruning the entire layer in a single pruning iteration, leading to performance degradation. On the other hand, small group sizes such as 2 promote removing unimportant groups of neurons. These groups do not significantly improve the latency, but they can contribute to the final performance. In contrast, our latency-aware grouping performs the best, showing the efficacy of our neuron grouping scheme.

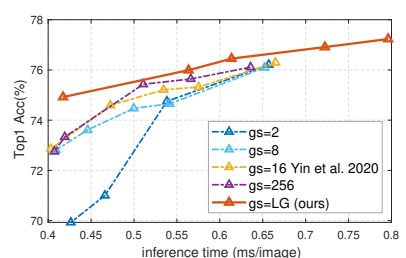

Figure 4: Performance comparison of our latency-aware grouping to different fixed sizes for ResNet50 pruning on ImageNet. We compare to heuristic-based group selection studied by Yin et al. (2020). LG denotes our proposed latency-aware grouping in HALP that yields consistent latency benefits per accuracy.

**Algorithm efficiency improvement.** Setting the group size according to the latency step size not only improves the performance, but also reduces computation cost on knapsack problem solving for neuron selection since it reduces the total number of object $N$ to a smaller value. In our ResNet50 experiment, except for the first convolution layer, the group size of other layers varies from 32 to 128. By neuron grouping, the value of $N$ can be reduced to 215, which takes around one minute on average at each pruning step to solve the knapsack problem on CPU. We have 30 pruning steps in total in our experiments, thus the time spent on neuron selection is around 30 minutes in total, which can be negligible compared to training time.

## 4.5 GENERALIZATION TO OBJECT DETECTION.

To show the generalization ability of our proposed HALP algorithm, we also apply the algorithm to the object detection task. In this experiment we take the popular architecture Single Shot Detector (SSD) Liu et al. (2016) on the PASCAL VOC dataset Everingham et al. (2010). Following the "07+12" setting in Liu et al. (2016), we use the union of VOC2007 and VOC2012 trainval as our training set and use the VOC2007 test as test set. We pretrain a SSD512 detector with ResNet50 as the backbone. The details of the SSD structure are elaborated in the appendix. Same to classification task, we prune the trained detector and finetune afterwards. We only prune the backbone network in the detector.

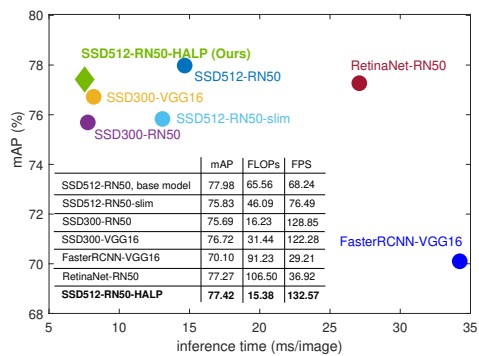

Figure 5: HALP for object detection on the PASCAL VOC dataset. Detailed numbers can be found in the Appendix D.

The results in Fig. 5 show that the pruned detectors maintain the similar final mAP but reduce the FLOPs and improve the inference speed greatly, with 77% FLOPs reduction and around $1.94\times$ speedup at the cost of only $0.56\%$ mAP drop. We compare the pruned detector to some other commonly-used detectors in the table. The results show that pruning a detector using HALP improves performance in almost all aspects.

## 5 CONCLUSION

We proposed hardware-aware latency pruning (HALP) that focuses on structured pruning for underlying hardware towards latency budgets. We formulated pruning as a resource allocation optimization problem to achieve maximum accuracy within a given latency budget. We further proposed a latency-aware neuron grouping scheme to further improve latency reduction. Over multiple neural network architectures, classification and detection tasks, and changing datasets, we have shown the efficiency and efficacy of HALP by showing consistent improvements over state-of-the-art methods.

## REPRODUCIBILITY STATEMENT

For reproducibility purpose, we have included all the experimental setup, details, and all hyperma-rameters in the paper and the Appendix. The code will be released upon acceptance.

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

## A    EXPERIMENTAL SETTINGS

For image classification, we focus on pruning networks on the large-scale ImageNet ILSVRC2012 dataset Russakovsky et al. (2015) (1.3M images, 1000 classes). Each pruning process consumes a single node with eight NVIDIA Tesla V100 GPUs. We use PyTorch Paszke et al. (2017) V1.4.0 model zoo for pretrained weights for our pruning for a fair comparison with literature.

In our experiments we perform iterative pruning. Specifically, we prune every 320 minibatches after loading the pretrained model with $k = 30$ pruning steps in total to satisfy the constraint. We finetune the network for 90 epochs in total with an individual batch size at 128 for each GPU. For finetuning, we follow NVIDIA's recipe Nvidia (2020) with mixed precision and Distributed Data Parallel training. The learning rate is warmed up linearly in the first 8 epochs and reaches the highest learning rate, then follows a cosine decay over the remaining epochs Loshchilov & Hutter (2017). For latency lookup table construction, we target a NVIDIA TITAN V GPU with batch size 256 for latency measurement to allow for highest throughput for inference, and target a Jetson TX2 with inference batch size 32. We pre-generate a layer latency look-up table on the platform by iteratively reducing of the number of neurons in a layer to characterize the latency with NVIDIA cuDNN Chetlur et al. (2014) V7.6.5. We profile each latency measurement 100 times and take the average to avoid randomness.

## B    EFFICACY OF NEURON GROUPING ON MOBILENET

In this section, we show the benefits of latency-aware neuron grouping and the performance under different group size settings on MobileNetV1.

Since MobileNet has group convolutional layers to speedup the inference, we take the group convolutional layer with its preceding connected convolutional layer together as coupled cross-layers Gao et al. (2019) to make sure the input channel number and output channel number of the group convolution remain the same. All the 27 convolutional layers can be divided into 14 coupled layers. In our method, with the neuron grouping, we set the individual group size of 1 coupled layer to 16, of 3 coupled layers to 32 and 10 coupled layers to 64. Also, for MobileNetV1 pruning, we add the additional constraint that each layer has at least one group of neurons remaining to make sure that the pruned network is trainable.

We compare our latency-aware neuron grouping with an heuristic option by setting a fixed group size for all layers. Fig. 6 shows the comparison results between our neuron grouping method and various fixed group sizes for a MobielNet pruned with different latency constraints on ImageNet. As shown, similar to ResNet50, using small group sizes such as 8, 16 leads to worse performance; a large group size like 128 also harms the performance significantly. Our observations on ResNet50 pruning also hold in MobileNetV1 setting, further emphasizing the efficacy of our latency-aware neuron grouping.

## C    COMPARISON WITH EAGLEEYE ON IMAGENET

We now use the same unpruned baseline model provided by EagleEye Li et al. (2020) to compare our proposed HALP method with EagleEye Li et al. (2020) varying the latency constraint. As shown in Fig. 7, our approach dominates EagleEye by consistently delivering a higher top-1 accuracy with a significantly faster inference time.

We then analyze the structure difference between our pruned model and the EagleEye model. As mentioned in the main text that the proposed HALP method tries to make the number of remaining neurons in each layer fall to the right side of a step if the latency on the targeting platform presenting a staircase pattern. Fig. 8 shows two examples of pruned layers after pruning from HALP-45% and EagleEye-2G model. In the left figure, we show that the layer in our pruned model has only 5 more neurons pruned than that in EagleEye model, the latency is reduced to a much lower level which is a 0.76ms drop while we have 31 more input channels. In the right figure, we also show that sometimes we can remain a lot more neurons (30 neurons) in layer with only little latency (0.21ms) increase. These two examples both show the ability of method to fully exploit the latency traits and benefit the inference speed.

| Model | mAP | FLOPs (G) | params (M) | FPS (BS=1) | FPS (BS=32) |
|---|---|---|---|---|---|
| SSD512-RN50, base model | 77.98 | 65.56 | 21.97 | 68.24 | 103.48 |
| SSD512-RN50-slim | 75.83 | 46.09 | 16.33 | 76.49 | 114.80 |
| SSD300-RN50 | 75.69 | 16.23 | 15.43 | 128.85 | 309.32 |
| SSD300-VGG16 Liu et al. (2016) | 76.72 | 31.44 | 26.29 | 122.28 | 262.93 |
| FasterRCNN-VGG16 Ren et al. (2015) | 70.10 | 91.23 | 137.08 | 29.21 | - |
| RetinaNet-RN50 Lin et al. (2017) | 77.27 | 106.50 | 36.50 | 36.92 | - |
| **SSD512-RN50-HALP (Ours)** | **77.42** | **15.38** | **10.40** | **132.57** | **323.36** |

Table 4: HALP for object detection on the PASCAL VOC dataset.

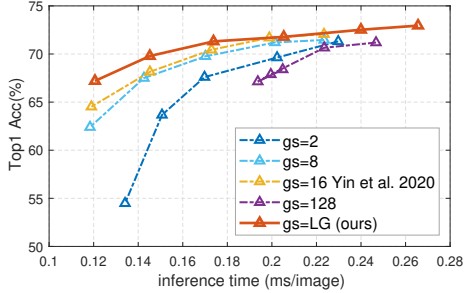 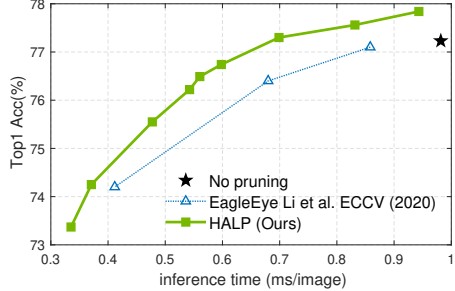

Figure 6: Performance comparison of our latency-aware grouping to different fixed sizes for a MobielNetV1 pruned with different latency constraints on ImageNet. We compare to heuristic-based group selection studied by Yin et al. (2020). LG denotes the proposed latency-aware grouping in HALP that yields consistent latency benefits per final accuracy.

Figure 7: Pruning ResNet50 on the ImageNet dataset using the same baseline model as in EagleEye with a top-1 accuracy of 77.23%. The proposed HALP surpasses EagleEye ECCV20 Li et al. (2020) in accuracy and latency. Top-left is better.

Our method benefits a lot from the non-linear latency characteristic since we are trying to keep as many neurons as possible under the latency constraint. If the latency of the layer on the targeting platform shows linear pattern, the advantage of our method becomes smaller. Fig. 8 shows the latency behavior of the example layers on the targeting platform when reducing the number of input and output channels. As we can see, the staircase pattern becomes less obvious as the number of input channel reduces and the GPU has sufficient capacity for the reduced computation. This happens during pruning, especially for large prune ratios. In such a case, the FLOP count reflects the latency more accurately, and the performance gap between reducing FLOPs and reducing latency can possibly become small. Nevertheless, our method can help avoid some latency peaks as shown in Fig. 8, which could otherwise happen using other pruning methods.

## D   PRUNING RESULTS ON OBJECT DETECTION

In this section we show the detailed pruning results on objection detection task for Sec. 4.5. To prune the detector, we first train a SSD512 with ResNet50 as backbone. We also train some other popular models for performance comparison. The detailed numbers of Fig. 5 are shown in Tab. 4.

## E   RESULTS WITH SMALL BATCH SIZE

In the main paper, we use a large batch 256 in the experiment to allow for highest throughput for inference, which also makes the latency of the convolution layers show apparent staircase pattern so that we can take full advantage of the latency characteristic. In this section, we show that with small batch size 1 that no obvious staircase pattern showing up in layer latency, our HALP algorithm still delivers better results compared to other methods.

When we use batch size 1 for inference, the layer latency of ResNet50 does not show obvious staircase pattern in most of the layers due to the insufficient usage of GPU. Therefore in this experiment, we use the latency lookup table granularity as a neuron grouping size, which in our case is 2, to fully

| Method | FLOPs (G) | Top1 Acc (%) | Top5 Acc (%) | FPS (imgs/s) | Speedup |
|---|---|---|---|---|---|
| No pruning | 4.1 | 76.2 | 92.87 | 181 | 1× |
| 0.75× ResNet50 He et al. (2015) | 2.3 | 74.8 | - | 192 | 1.06× |
| AutoSlim Yu & Huang (2019) | 2.0 | 75.6 | - | 181 | 1.00× |
| MetaPruning Liu et al. (2019a) | 2.0 | 75.4 | - | 190 | 1.05× |
| EagleEye-2G Li et al. (2020) | 2.1 | 76.4 | 92.89 | 190 | 1.05× |
| GReg-2 Wang et al. (2021) | 1.8 | 75.4 | - | 196 | 1.09× |
| **HALP**-90% **(Ours)** | 2.9 | **76.4** | **93.10** | **220** | **1.22×** |
| 0.50× ResNet50 He et al. (2015) | 1.1 | 72.0 | - | 193 | 1.07× |
| AutoSlim Yu & Huang (2019) | 1.0 | 74.0 | - | 191 | 1.06× |
| MetaPruning Liu et al. (2019a) | 1.0 | 73.4 | - | 196 | 1.09× |
| EagleEye-1G Li et al. (2020) | 1.0 | 74.2 | 91.77 | 192 | 1.06× |
| GReg-2 Wang et al. (2021) | 1.3 | 73.9 | - | 206 | 1.14× |
| **HALP**-80% **(Ours)** | 2.3 | **75.3** | **92.35** | **247** | **1.37×** |

Table 5: Pruning ResNet50 on the ImageNet dataset (TITAN V) targeting on inference with batch size 1. HALP-$X$% indicates that $X$% latency to remain after pruning. The speedup is calculated as the ratio of FPS between the pruned network and the unpruned model.

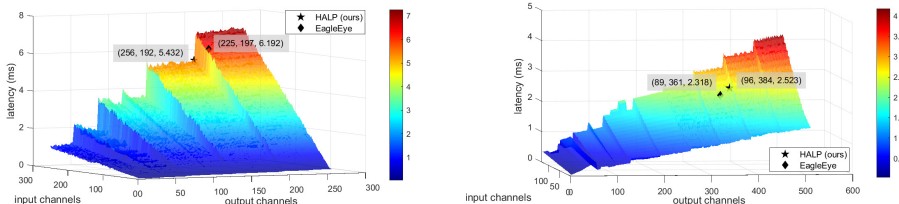

Figure 8: Two examples of pruned layers from HALP model and EagleEye Li et al. (2020) model. The scattered black points are the locations of the layers fall to after pruning.

exploit the hardware latency traits during pruning. We show our pruned results and the comparison with other methods in Tab. 5. As shown in the table, while other methods reduce the total FLOPs of the network after pruning, they do not reduce the actual latency much, which is up to 1.09× faster than the original one at the cost of 2.8% top1 accuracy drop. Compared to these methods, although we get less FLOPs reduction using our proposed method, the pruned models are faster and get higher accuracy, which is 1.22× faster than the unpruned model while getting slightly higher accuracy and 1.37× faster with only 0.9% accuracy drop.

## F  IMPLEMENTATION DETAILS

**Convert latency in float to int.** Solving the neuron selection problem using the proposed augmented knapsack solver (Algo. 1 in the main paper), requires the neuron latency contribution and the latency constraint to be integers as shown in line 4 of the algorithm. To convert the measured latency from a full precision floating-point number to integer type, we multiply the latency by 1000 and perform rounding. Accordingly, we also scale and round the latency constraint value.

**Deal with negative latency contribution.** The neuron latency contribution in our augmented knapsack solver must be a non-negative value since we have dp_array $\in \mathbb{R}^C$ and we need to visit dp_array$[c - c_n]$ as in line 5 of Algo. 1 in the main paper. However, by analyzing the layer latency from the look-up table we find that for some layers the measured latency might even increase when reducing some number of neurons. This means that the latency contribution could possibly be negative. The simplest way to deal with the negative values is to directly set the negative latency contributions to be 0. This leads to the problem that the summed latency contribution would be larger than the actual latency value, causing less neurons being selected. Thus, during our implementation, we keep those negative latency values as they are, but update the vector size of dp_array

to $\mathbb{R}^{C-\min(\min(\mathbf{c}),0)}$ where $\min(\mathbf{c})$ is the minimum latency contribution. With such, the vector size of dp_array would be extended when there is negative latency contribution. This makes it possible to add one neuron with negative latency contribution to a subset of neurons whose summed latency is larger than the latency constraint. After the addition, the total latency will still remain under the constraint.

**Pruning of the first layer.** In our experiments, we leave the first convolutional layer of ResNets unpruned to help maintain the top-1 classification accuracy. For MobileNet, the first convolutional layer is coupled with its following group convolutional layer. In our MobileNet experiments, we prune the first coupled layers at most to the half of neurons.

**SSD for object detection.** Our SSD model is based on Liu et al. (2016). When we train SSD-VGG16, we use the exactly same structure as described in the paper. When we train a SSD-ResNet50, the main difference between our model and the model described in the original paper is in the backbone, where the VGG is replaced by the ResNet50. Following Huang et al. (2017), we apply the following enhancements in our backbone:

- The last stage of convolution layers, last avgpool and fc layers are removed from the original ResNet50 classification model.
- All strides in the 3rd stage of ResNet50 layers are set to $1 \times 1$.

The backbone is followed by 6 additional coupled convolution layers for input size $512 \times 512$, or 5 for input size $300 \times 300$. A BatchNorm layer is added after each convolution layer. The settings of these additional convolution layers are listed in Tab. 6, each layer is represented as (output channel, kernel size, stride, padding).

| layer | SSD512 | SSD512-slim | SSD300 |
|---|---|---|---|
| layer1-conv1 | (512, 3, 1, 1) | (256, 1, 1, 0) | (512, 3, 1, 1) |
| layer1-conv2 | (512, 3, 2, 1) | (512, 3, 2, 1) | (512, 3, 2, 1) |
| layer2-conv1 | (256, 1, 1, 0) | (256, 1, 1, 0) | (256, 1, 1, 0) |
| layer2-conv2 | (512, 3, 2, 1) | (512, 3, 2, 1) | (512, 3, 2, 1) |
| layer3-conv1 | (128, 1, 1, 0) | (128, 1, 1, 0) | (128, 1, 1, 0) |
| layer3-conv2 | (256, 3, 2, 1) | (256, 3, 2, 1) | (256, 3, 2, 1) |
| layer4-conv1 | (128, 1, 1, 0) | (128, 1, 1, 0) | (128, 1, 1, 0) |
| layer4-conv2 | (256, 3, 2, 1) | (256, 3, 2, 1) | (256, 3, 1, 0) |
| layer5-conv1 | (128, 1, 1, 0) | (128, 1, 1, 0) | (128, 1, 1, 0) |
| layer5-conv2 | (256, 3, 2, 1) | (256, 3, 2, 1) | (256, 3, 1, 0) |
| layer6-conv1 | (128, 1, 1, 0) | (128, 1, 1, 0) | - |
| layer6-conv2 | (256, 4, 1, 1) | (256, 4, 1, 1) | - |

Table 6: The additional convolution layers in SSD.

The detector heads are similar to the ones in the original paper. The first detection head is attached to the last layer of the backbone.The rest detection heads are attached to the corresponding additional layers. No additional BatchNorm layer in the detector heads.

## G FLOPs-constrained Pruning

Our implementation of latency-constrained pruning can be easily converted to be a FLOPs-constrained. When constraining on FLOPs, $\Phi(\cdot)$ in the objective function (Eq.1 in the main paper) becomes the FLOPs measurement function and $C$ becomes the FLOPs constraint. Since the FLOPs of a layer linearly decreases as the number of neurons decreases in the layer, we do not need to group neurons in a layer any more. The problem can also be solved by original knapsack solver since each neuron's FLOPs contribution in a layer is exactly the same and no preceding constraint is required. We conduct some experiments by constraining the FLOPs and compare the results with EagleEye Li et al. (2020). We name the experiments using the same algorithm as HALP but targeting on optimizing the FLOPs as FLOP-T. As shown in Tab. 7, with our pruning framework applying the knapsack solver, our results show higher top-1 accuracy compared to the pruned networks of EagleEye with

| Method | FLOPs (G) | Top1 Acc (%) | Top5 Acc (%) |
|---|---|---|---|
| No pruning | 4.1 | 77.23 | 93.70 |
| EagleEye-3G | 3.08 | 77.10 | 93.36 |
| **FLOP-T (Ours)** | **2.99** | **77.36** | **93.62** |
| EagleEye-2G | 2.06 | 76.38 | 92.90 |
| **FLOP-T (Ours)** | **1.95** | **76.64** | **93.21** |
| EagleEye-1G | 1.03 | 74.18 | 91.78 |
| **FLOP-T (Ours)** | **0.96** | **74.84** | **92.26** |

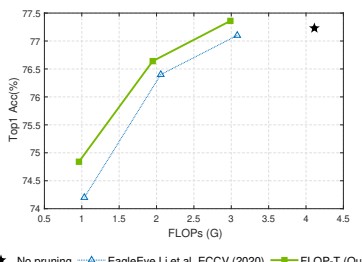

Table 7: Pruning ResNet50 on the ImageNet dataset with FLOPs constraint and comparison with state-of-the-art method EagleEye (ECCV'20) Li et al. (2020). We remeasure the FLOPs, top1 and top5 accuracy of EagleEye to get results with two digits.

similar FLOPs remaining. We also observe a larger gap between the methods when it comes to a more compact network.

## H    FLOPS VS. LATENCY

FLOPs can be regarded as a proxy of inference latency; however, they are not equivalent [4, 33, 35, 40, 42]. We do global filter-wise pruning and have the same problem as NAS. The latency on a GPU usually imposes staircase-shaped patterns for convolutional operators with varying channels and requires pruning in groups. In contrast, FLOPs will change linearly. Depth-wise convolution, compared to dense counterparts, has significantly fewer FLOPs but almost the same GPU latency due to execution being memory-bounded[1]. The discrepancy also holds for ResNets where the same amount of FLOPs impose more latency in earlier layers than later ones as the number of channels increases and feature map dimension shrinks – both increase compute parallelism. For example, the first $7 \times 7$ conv layer and the first bottleneck $3 \times 3$ conv in ResNet50 have nearly identical FLOPs but the former is $60\%$ slower on-chip.

We compare our results of FLOPs-targeted (FLOP-T) showed in Sec. G and results using latency-targeted pruning (HALP) in Tab. 8. As shown in the table, using different optimization targets leads to quite different FPS vs FLOPs curves. In overall, with similar FLOPs remaining, using our HALP algorithm targeting on reducing the actual latency can get more efficient networks with more image being processed per second.

| method | Top1(%) | FLOPs (G) | FPS (imgs/s) | FPS vs FLOPs | | |
|---|---|---|---|---|---|---|
| FLOP-T | 74.84 | **0.962** | 2202 | | | |
| HALP | **74.92** | 1.210 | **2396** | | | |
| FLOP-T | **76.64** | **1.949** | 1436 | | 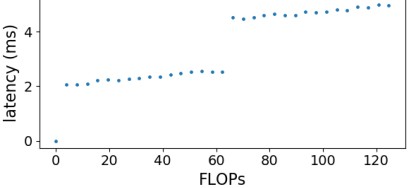 | |
| HALP | 76.55 | 1.957 | **1672** | | | |
| FLOP-T | 77.36 | **2.988** | 1146 | | | |
| HALP | **77.45** | 2.988 | **1203** | | | |

Table 8: ResNet50 pruning with FLOPs/latency constrain.

We also show a more straightforward relationship between the actual latency of a layer and its FLOPs in Fig. 9. We use the 2nd convolution layer in the 1st residual block of ResNet50 as an example. We vary the number of neurons of the layer from 0 to 128 and measure the actual latency on GPU (TITAN V) as well as the FLOPs of the layer. We can see from the figure that the actual latency does not strictly linearly decrease as the FLOPs decreasing.

Figure 9: The measured latency vs. FLOPs of the 2nd convolution layer in the 1st residual block of ResNet50.

---

[1] https://tlkh.dev/depsep-convs-perf-investigations/

## I   ALGORITHM 1 EXPLANATION

We follow the standard Knapsack problem dynamic programming solution to break down the original problem into sub-problems. Specifically, for each neuron $j$ (or neuron group) in layer $l$, we can choose to either include or not include it under the latency constraint $c$. When it is kept, the total importance score increases $\mathcal{I}_l^j$ while the latency constraint for the other neurons becomes $c - c_l^j$; If the neuron is removed, the latency constraint for the other neurons remains $c$. We choose to keep or remove the current neuron to maximize total importance. At the same time, we check whether the more important neurons in the same layer are included to ensure the correctness of the latency. The neuron selection from the remaining neurons is a sub-problem to solve.

Precisely, we use a vector maxV $\in \mathbb{R}^{(C+1)}$ to store the maximum importance that we can achieve under the latency constraint $c$, $0 \le c \le C$ and *keep* $\in \mathbb{R}^{L \times (C+1)}$, a 2D vector where keep$[l, c]$ denotes the number of neuron groups we need to maintain in layer $l$ to obtain the maximum importance maxV$[c]$. We process the neurons according to their importance score in decreasing order. In this way, all preceding neurons to the current one (i.e., neurons with a higher importance score in the same layer) will be always considered first. To decide if we keep or remove the current neuron, we check the total importance score and the inclusion status of its preceding neurons, so we can maximize the total importance and ensure the latency cost correctness.

## J   ABLATION STUDY OF PRUNING STEP $k$

In this work, similar to many other prior methods Alvarez & Salzmann (2016); Molchanov et al. (2019); Yu & Huang (2019), we do iterative pruning with $k$ pruning steps in total. In this experiment, we analyze the the accuracy of the final result as a function of $k$. We set the value of $k$ to 10, 20, 30 and 40 for iterative pruning, and also use $k = 1$ to perform a single-shot pruning. The result of this experiment is shown in Fig. 10. As shown, we get similar results independent of $k$. Imporantly, all these results outperform EagleEye Li et al. (2020). As expected, there is a drop in accu-

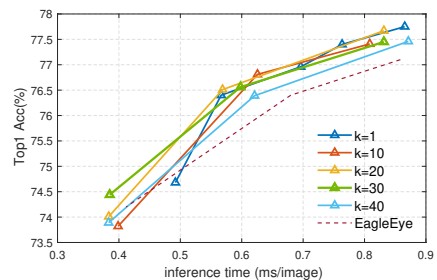

Figure 10: Performance comparison of different pruning steps $k$ for ResNet50 pruning on ImageNet.

racy for single-shot pruning ($k = 1$), especially for large pruning ratios. The main reason is the neuron importance would change as we remove some other neurons and, in this setting, the value is not updated. Iterative pruning does not have this limitation as the importance score and the latency cost of the remaining neurons is updated after each pruning step to reflect any changes. In our experiments, we use $k = 30$ as it provides a good trade off between latency and accuracy.

## K   MORE PRUNING RESULTS ON MOBILENETS

Table. 9 and Fig. 11 provide additional pruning results for lightweight networks such as MobileNet-V1 and MobileNet-V2. For the unpruned models, we find that even MobileNet-V2 has significantly lower FLOPs, the inference time is larger compared to MobileNet-V1. In both cases, HALP yields inference speeds-ups of $1.22\times$ and $1.33\times$ for MobileNet-V1 and MobileNet-V2 respectively, while maintaining the original top1 accuracy.

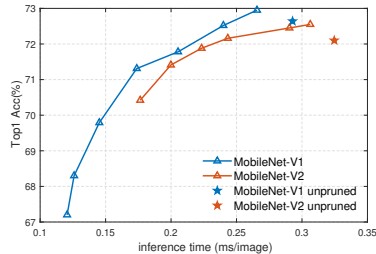

Figure 11: Pruning MobileNets on the ImageNet dataset.

| Method | FLOPs (M) | Top1 (%) | Top5 (%) | FPS (im/s) | Speedup | Method | FLOPs (M) | Top1 (%) | Top5 (%) | FPS (im/s) | Speedup |
|---|---|---|---|---|---|---|---|---|---|---|---|
| **MobileNet-V1** | | | | | | **MobileNet-V2** | | | | | |
| No pruning | 569 | 72.64 | 90.88 | 3415 | 1× | No pruning | 301 | 72.10 | 90.60 | 3080 | 1× |
| HALP-40% | 154 | 67.20 | 87.32 | 8293 | 2.43× | HALP-60% | 183 | 70.42 | 89.75 | 5668 | 1.84× |
| HALP-42% | 171 | 68.30 | 88.08 | 7940 | 2.32× | HALP-65% | 218 | 71.41 | 90.08 | 5003 | 1.62× |
| HALP-50% | 237 | 69.79 | 89.08 | 6887 | 2.02× | HALP-70% | 227 | 71.88 | 90.39 | 4478 | 1.45× |
| HALP-60% | 297 | 71.31 | 90.05 | 5754 | 1.68× | HALP-75% | 249 | 72.16 | 90.44 | 4109 | 1.33× |
| HALP-70% | 360 | 71.78 | 90.39 | 4870 | 1.43× | HALP-90% | 273 | 72.45 | 90.68 | 3443 | 1.12× |
| HALP-80% | 416 | 72.52 | 90.78 | 4167 | 1.22× | HALP-95% | 281 | 72.55 | 90.79 | 3265 | 1.06× |
| HALP-90% | 507 | 72.95 | 91.02 | 3765 | 1.10× | | | | | | |

Table 9: Pruning MobileNet-V1 and MobileNet-V2 on the ImageNet dataset with different targets.

## L DETAILED CONFIGURATION OF PRUNED MODELS

We provide the detailed configuration of our pruned models of Tab. 1. For each model, we list the number of neurons remaining in each convolution layer, starting from the input to the output. For ResNets we use [·] to denote a residual block and (·) to denote the neuron number of the residual bypass layer.

Note that for ResNets, the configuration is the "raw" configuration after the pruning. For each residual block $[x_1, x_2, x_3]$, because of the existing of bypass layer, we allow the entire layer pruning and it still leads to trainable networks. Whenever there is entire layer pruning ($x_1 == 0$ or $x_2 == 0$ or $x_3 == 0$), all the other layers in the block can be removed and this branch can be replaced by a constant value. In such a case, the corresponding block in our final pruned model is cleaned to be $[0, 0, 0]$. We also provide the detailed structure plot of two pruned ResNet50 models in Fig. 12 for a better visualization.

| | **ResNet50 - EagleEye Li et al. (2020) baseline** |
|---|---|
| HALP-80% | 64, [64, 32, 256](256), [32, 32, 256], [0, 32, 256], [128, 128, 512](512), [64, 96, 512], [64, 128, 512], [64, 128, 512], [256, 256, 1024](1024), [256, 160, 1024], [256, 160, 1024], [256, 160, 1024], [256, 128, 1024], [256, 160, 1024], [512, 512, 2048](2048), [448, 416, 2048], [512, 512, 2048] |
| HALP-45% | 64, [64, 32, 128](128), [32, 0, 128], [0, 32, 128], [64, 64, 384](384), [64, 96, 384], [32, 96, 384], [64, 128, 384], [256, 192, 1024](1024), [128, 96, 1024], [256, 64, 1024], [256, 96, 1024], [128, 32, 1024], [256, 96, 1024], [512, 448, 2048](2048), [416, 288, 2048], [512, 352, 2048] |
| HALP-30% | 64, [0, 0, 64](64), [0, 0, 64], [0, 32, 64], [64, 64, 256](256), [64, 64, 256], [64, 64, 256], [64, 96, 256], [256, 64, 896](896), [128, 64, 896], [0, 0, 896], [128, 64, 896], [0, 0, 896], [0, 0, 896], [512, 320, 2048](2048), [320, 160, 2048], [480, 160, 2048] |
| | **ResNet101** |
| HALP-60% | 64, [64, 32, 192](192), [32, 32, 192], [64, 32, 192], [64, 128, 384](384), [64, 96, 384], [96, 96, 384], [128, 128, 384], [256, 256, 896](896), [256, 96, 896], [256, 128, 896], [256, 96, 896], [256, 96, 896], [256, 128, 896], [256, 96, 896], [256, 96, 896], [256, 96, 896], [256, 96, 896], [256, 64, 896], [256, 64, 896], [256, 96, 896], [256, 64, 896], [256, 64, 896], [256, 96, 896], [256, 64, 896], [256, 96, 896], [256, 96, 896], [256, 96, 896], [256, 64, 896], [256, 128, 896], [256, 64, 896], [512, 512, 2048](2048), [512, 448, 2048], [512, 480, 2048] |
| HALP-50% | 64, [64, 32, 128](128), [0, 32, 128], [64, 32, 128], [64, 128, 384](384), [32, 32, 384], [128, 96, 384], [128, 96, 384], [256, 256, 896](896), [256, 96, 896], [256, 96, 896], [256, 96, 896], [256, 64, 896], [256, 96, 896], [256, 95, 896], [256, 96, 896], [256, 64, 896], [256, 64, 896], [0, 0, 896], [0, 0, 896], [256, 64, 896], [0, 0, 896], [256, 64, 896], [256, 64, 896], [256, 64, 896], [256, 64, 896], [256, 64, 896], [256, 64, 896], [512, 512, 2048](2048), [480, 416, 2048], [512, 416, 2048] |
| HALP-40% | 64, [64, 32, 128](128), [0, 32, 128], [0, 0, 128], [64, 128, 384](384), [32, 64, 384], [64, 96, 384], [64, 128, 384], [256, 256, 896](896), [0, 64, 896], [128, 96, 896], [0, 64, 896], [0, 64, 896], [128, 64, 896], [128, 64, 896], [0, 0, 896], [0, 0, 896], [128, 64, 896], [128, 96, 896], [256, 64, 896], [256, 64, 896], [256, 96, 896], [256, 64, 896], [256, 96, 896], [128, 32, 896], [128, 64, 896], [0, 0, 896], [0, 0, 896], [256, 64, 896], [256, 64, 896], [256, 96, 896], [256, 64, 896], [256, 96, 896], [512, 512, 2048](2048), [512, 352, 2048], [512, 352, 2048] |
| HALP-30% | 64, [64, 32, 128](128), [0, 0, 128], [0, 0, 128], [64, 128, 256](256), [0, 96, 256], [64, 96, 256], [64, 128, 256], [256, 192, 768](768), [0, 64, 768], [128, 64, 768], [0, 64, 768], [0, 64, 768], [128, 64, 768], [128, 64, 768], [0, 0, 768], [0, 0, 768], [128, 64, 768], [128, 96, 768], [256, 64, 768], [0, 0, 768], [0, 0, 768], [128, 32, 768], [128, 64, 768], [128, 64, 768], [128, 64, 768], [0, 0, 768], [0, 0, 768], [256, 64, 768], [256, 0, 768], [256, 64, 768], [512, 512, 2048](2048), [480, 288, 2048], [480, 256, 2048] |
| | **MobileNet-V1** |
| HALP-60% | 896, 14, 14, 32, 32, 64, 64, 64, 64, 192, 192, 192, 192, 384, 384, 320, 320, 384, 384, 384, 384, 384, 384, 448, 448, 960, 960 |
| HALP-42% | 832, 16, 16, 32, 32, 32, 32, 32, 32, 64, 64, 128, 128, 320, 320, 256, 256, 256, 256, 256, 256, 320, 320, 320, 320, 896, 896 |
| | **MobileNet-V2** |
| HALP-75% | 16, 16, 16, 64, 64, 24, 64, 64, 24, 112, 112, 32, 128, 128, 32, 128, 128, 32, 192, 192, 64, 384, 384, 64, 352, 352, 64, 352, 352, 64, 384, 384, 96, 512, 512, 96, 512, 512, 96, 576, 576, 160, 960, 960, 160, 960, 960, 160, 960, 960, 320, 1280 |
| HALP-60% | 16, 16, 8, 32, 32, 16, 16, 16, 16, 64, 64, 32, 32, 32, 32, 64, 64, 32, 176, 176, 64, 288, 288, 64, 320, 320, 64, 320, 320, 64, 384, 384, 96, 448, 448, 96, 448, 448, 96, 576, 576, 160, 960, 960, 160, 960, 960, 160, 960, 960, 192, 1152 |

Table 10: The detailed configuration of the HALP pruned models.

## M DIFFERENT CHOICE OF IMPORTANCE CALCULATION

We use the first Taylor expansion Molchanov et al. (2019) to estimate the loss change induced by pruning as the importance score of the neurons. It is a gradient-based importance calculation and is shown to be given promising results. In this section, we use the L2 norm of the neuron weights as the importance measurement and apply HALP framework to ResNet50 ImageNet classification

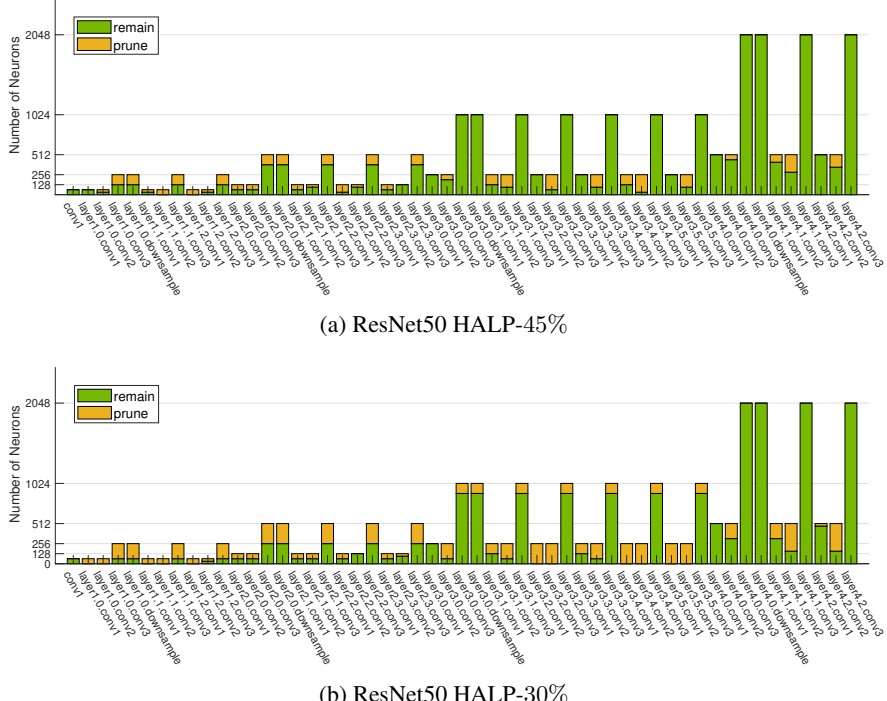

(a) ResNet50 HALP-45%

(b) ResNet50 HALP-30%

Figure 12: Visualization of the pruned ResNet50 structure.

task. As shown in Tab. 11, Our algorithm is generic applying to different importance measurements. As shown, using L2 norm of weights as importance measurements leads to slightly lower accuracy.

| Method | First-Taylor Expansion (gradient-based) | | | | L2 norm (magnitude-based) | | | |
|--------|-----------|---------|---------|----------|-----------|---------|---------|----------|
| | FLOPs(G) | Top1(%) | Top5(%) | FPS(im/s) | FLOPs(G) | Top1(%) | Top5(%) | FPS(im/s) |
| HALP-80% | 3.0 | 77.5 | 93.60 | 1203 | 3.0 | 77.3 | 93.60 | 1196 |
| HALP-55% | 2.1 | 76.7 | 93.16 | 1672 | 2.0 | 75.7 | 92.66 | 1595 |

Table 11: The results of HALP algorithm on ResNet50 ImageNet classification task with different choices of neuron importance measurements.

## N  LATENCY LOOK-UP TABLE CREATION AND CALIBRATION

In this section, we provide additional details to build the latency look-up table used in HALP, computational cost and the correlation between the estimated and the real ones. As mentioned in Appendix A, we pre-generate the layer latency look-up table on the platform with NVIDIA cuDNN Chetlur et al. (2014) V7.6.5. For each layer, we iteratively reduce the number of neurons in the layer (each time reduce 8 neurons) and characterize the corresponding latency. For each latency measurement, we use one profile for GPU warm up and another 3 profiles and take the average to avoid randomness. The average standard deviation of profiles for an operation is $8.67e^{-3}$.

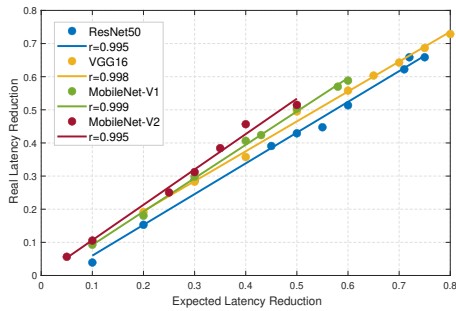

Figure 13: The correlation between the predicted latency reduction and the real latency reduction of the pruned models.

On a single TITAN V GPU, it takes around 5 hours to build the look-up table for ResNets family and 1 hour for MobileNets family. Note that the LUT can be shared by the network architectures within the same family since they usually have similar layer structures. We only need to create the latency look-up table once for all the possible latency targets.

There are indeed some gaps between the predicted latency and the real latency of the model, because the latency look-up table is created layer-wise on convolution layers. There are additional costs in real inference such as pooling, non-linear activation etc. We plot the correlation between the expected latency reduction from look-up table and the real latency reduction ratio of our pruned models in Fig. 13. We also calculate the Pearson Correlation Coefficient $r$ for all the networks in the figure. As shown, we can clearly see a linear correlation between the predicted and real latency reduction.

## O    BREAKDOWN OF THE ALGORITHM EXECUTION TIME

We provide additional details of the algorithm process of different methods in this section. We estimate the time cost needed to get the pruned network structure for each method. The time of following finetuning is not taken into consideration. For a fair comparison, we set the number of pruning steps $k$ for all iterative pruning methods to 30. All the values are approximated as all the methods are running on the same device (a NVIDIA V100 GPU) to get a pruned ResNet50. For AutoSlim Yu & Huang (2019), MetaPruning Liu et al. (2019a) and AMC He et al. (2018), more GPU time is needed for additional training of the network.

| Method | Evaluate proposals? | Auxiliary net training? | Sub-network selection | Additional time cost | Estimate time (RN50) |
|---|---|---|---|---|---|
| NetAdapt | Y | N | $N$ candidates evaluation + finetune after each prune. Repeat $k$ times | Latency look-up table creation | $\sim$ 195h GPU |
| ThiNet | Y | N | 1 or 2 train epochs after each pruning. Repeat $k$ times | Additional forward pass to get neuron importance | $\sim$ 210h GPU |
| EagleEye | Y | N | 1000 candidates evaluation | Monte Carlo sampling, prune to get 1000 candidates | 30h GPU |
| AutoSlim | Y | Y | Train slimmable model $k$ candidates evaluation | | |
| MetaPruning | Y | Y | Train an auxiliary network $k$ candidates evaluation | | |
| AMC | N | Y | Train an RL agent | | |
| HALP(**Ours**) | **N** | **N** | 40 train iterations after each pruning. Repeat 30 times. ($<$ 1 train epoch in total) | Augmented Knapsack solver ($\sim$ 30min in total) Latency look-up table creation | 6.5h GPU 0.5h CPU |

Table 12: The breakdown details of the execution process of different methods.

## P    PRUNING TARGETING OTHER HARDWARE

In this section, we provide additional experiments targeting two additional hardware platforms: Intel CPU Xeon E5 and NVIDIA Xavier. In particular, we generate look up tables for those two hardware and compare results to the state-of-the-art EagleEye Li et al. (2020). For inference on CPU, we use a batch size of 8; for NVIDIA Xavier, we use batch size 128. Results for this experiment are show in Fig. 14.

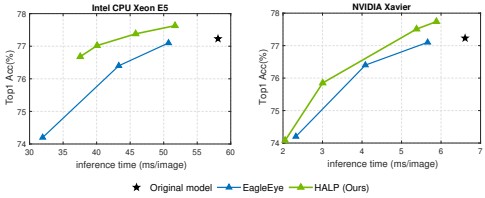

Figure 14: The ResNet50 ImageNet pruning targeting inference on different hardware.

As we can see in Fig. 14, our approach consistently outperforms EagleEye, demonstrating the benefits of considering the target hardware when performing pruning. HALP leverages the latency characteristics of the target hardware to make a better trade-off between the accuracy and efficiency. Specifically, compared to the original model on the Intel CPU, HALP yields $1.27\times$ speedup with a $0.15\%$ increment in top-1 accuracy; On the NVIDIA Xavier platform, HALP yields a $1.23\times$ speedup with a $0.28\%$ increment in top-1 accuracy. Compared to EagleEye, HALP achieves up to $1.45\times$ speedup with a similar accuracy. From these results, we can conclude that our method generalizes well to different hardware.

## Q    Pruning for INT8 Quantization

We now focus on results when the target is INT8 inference which is a common requirement for real-world applications. In particular, we use NVIDIA Xavier as the target platform as, in this platform, INT8 speedup is supported. We create a INT8 latency look-up table for INT8 inference. For comparison, we also create a FP32 latency look-up table on the same platform and use both look-up tables for pruning a ResNet50 model on ImageNet classification. After convergence, results are quantized into INT8 and the latency and accuracy is measured directly on the Xavier platform. We measure latency with a batch size 128, using TensorRT (V8.2.0.1) to get INT8 speedup. Results for this experiment are shown in Fig. 15.

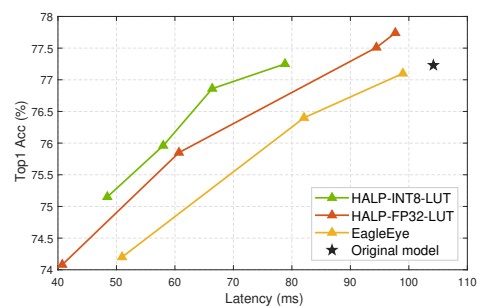

Figure 15: The ResNet50 ImageNet pruning targeting INT8 inference on NVIDIA Xavier.

As shown, even use a FP32 latency look-up table as the latency guidance, our method HALP outperforms the state-of-the-art method EagleEye Li et al. (2020). These results are consistent with Sec. 4.2 where we show the HALP acceleration on GPUs with TensorRT. Pruning results of using a INT8 look-up table show that our method yields higher accuracy with lower latency on this platform. We obtain a $1.32\times$ speedup while maintaining the original Top1 accuracy. Compared to EagleEye, HALP achieves up to $1.26\times$ relative speedup and $0.15\%$ higher accuracy.

