# OpenReview forum: "HALP: Hardware-Aware Latency Pruning"
_ICLR.cc/2022/Conference — ICLR 2022 Submitted_

### Official Review · Reviewer_ywGD · 2021-10-26

**Correctness:** 3
**Technical Novelty And Significance:** 3
**Empirical Novelty And Significance:** 3
**Recommendation:** 6
**Confidence:** 5

**Main Review:**

**Contribution**:

1. The authors formulate structural filter pruning as a global resource allocation optimization problem with latency constraint rather than theoretical computation cost (FLOPs).
2. The authors devise an augmented knapsack to solve the resulting combinatorial optimization problem.
3. Extensive experiments on image classification and object detection tasks show the promising performance of the proposed method.

**Questions and points needed to be improved**:

1. Algorithm 1 is confusing. Many notations are unclear and not rigorous. For example, what does dp_array denote? What does v_{keep} denote? Moreover, the authors do not provide any explanations regarding Algorithm 1, which makes it hard to follow. More explanations are required.

2. In Section 3.3, the authors state that the proposed method groups the neurons sharing the same channel index from the connected layers in a network with skip connections, which have been proposed in [1-2]. It would be better for the authors to cite the related papers.

3. In Section 3.4, the authors propose to perform pruning for k steps in total, which would improve the performance of the pruned models. It is unclear whether the good performance is resulting from multi-step pruning. It would be better for the authors to provide more ablation studies on the effect of different k.

4. What will happen if the authors replace the latency constraints with FLOPs constraints? Does the performance improvement of the proposed method come from a better metric of computation cost or a better solver? More discussion and experiments are required.

5. The authors conduct experiments on heavyweight networks (e.g., ResNet-50, ResNet-101, VGG-16) and a lightweight network (e.g., MobileNetV1). It would be more convincing for the authors to provide more results on lightweight networks with residual connections (e.g., MobileNetV2 [3]).

6. To demonstrate the effectiveness of the proposed method, it would be better for the authors to provide the detailed configurations (the number of channels of each layer) of the pruned models, which will strengthen the paper.

Minor issues:
1. Figure 1 is too small to read.

**References**:

[1] Centripetal sgd for pruning very deep convolutional networks with complicated structure. CVPR 2019.

[2] Neural Network Pruning with Residual-Connections and Limited-Data. CVPR 2020.

[3] MobileNetV2: Inverted Residuals and Linear Bottlenecks. CVPR 2018.

**Summary Of The Paper:**

The authors propose to formulate structural filter pruning as a global resource allocation optimization problem with latency constraints. To solve the resulting knapsack problem, the authors devise an augmented knapsack solver and further propose neuron grouping to reduce the pruning space and computational cost. Extensive experiments on image classification and object detection tasks show the promising performance of the proposed method.


**Summary Of The Review:**

The performance is promising. However, some important details of the proposed method are missing, which makes it hard to follow. Moreover, experiments are not sufficient. More experiments are required to show the effectiveness of the proposed method.

---

> ### Author Response · Authors · 2021-11-18
> **Rebuttal by paper723 Authors**
>
> Dear Reviewer ywGD,
>
> We would like to thank the reviewer for the valuable comments and suggestions. Below we provide clarifications to your comments. We have also integrated these comments into the main paper and appendixes (see paper update).
>
> C1. Algorithm 1 is confusing. Many notations are unclear and not rigorous. For example, what does dp_array denote? What does v_{keep} denote? Moreover, the authors do not provide any explanations regarding Algorithm 1, which makes it hard to follow. More explanations are required.
>
> We thank the reviewer for the suggestion to help improve the clarity of the algorithm. We have updated Algo.1 to improve clarity and make it more rigorous, including an easier connection to Eq.7. We have also added additional comments to the algorithm and a detailed explanation in Appendix I.
>
> C2. In Section 3.3, the authors state that the proposed method groups the neurons sharing the same channel index from the connected layers in a network with skip connections, which have been proposed in [1-2]. It would be better for the authors to cite the related papers.
>
> We thank the reviewer for pointing this out. We have now added the citation in Section 3.3. “When dealing with skip connections in ResNet and group convolutions in MobielNet, we not only group neurons within a layer, we also group the neurons sharing the same channel index from the connected layers Ding et al. (2019); Luo & Wu (2020).”
>
>
> C3. It would be better for the authors to provide more ablation studies on the effect of different k.
>
> We have added an ablation experiment to show the accuracy as a function of the pruning step k in Appendix J. We show results for different values of k (k=1, 10, 20, 30, 40) where k=1 represents single shot pruning, and compare these results to iterative pruning [1,2,3]. As shown in Fig. 10, our approach consistently yields better results compared to the state-of-the-art, independently of the value of k. For small prune ratios, single-shot pruning results in a competitive performance compared to iterative pruning (top right curve). However, for larger pruning ratios, there is a larger accuracy drop as, once a neuron is removed, the importance of neurons in that layer is no longer updated. As shown, using k=30 yields a good trade-off between accuracy and efficiency. Therefore, as also happens in [2], we choose k=30 for the experiments in our paper.
>
> C4. What will happen if the authors replace the latency constraints with FLOPs constraints? Does the performance improvement of the proposed method come from a better metric of computation cost or a better solver? More discussion and experiments are required.
>
> Thank you for pointing this out. The experiment using FLOPS as a constraint was already included in the supplemental material in Appendix G. In that experiment, we compared our results to existing FLOP-based methods. As shown in Tab.7, with FLOPs-constrained pruning, our algorithm outperforms the state of the art [4]. Our approach leads to fewer FLOPs and higher accuracy. These results demonstrate the generalizability of our proposed algorithm.
>
>
> C5. It would be more convincing for the authors to provide more results on lightweight networks with residual connections (e.g., MobileNetV2).
>
> As requested by the reviewer, we conducted one experiment pruning MobileNetV2. The results show that our proposed HALP algorithm also works on lightweight networks with residual connections. For MobileNetV2, for the same accuracy, our approach yields 1.33X speedup (ours: 72.16% / 4109 fps; MobileNetV2: 72.10% / 3080 FPS). We have added these numbers in Tab.1 of the main paper and in Figure 11 and Tab.9 in appendix K.
>
> C6. To demonstrate the effectiveness of the proposed method, it would be better for the authors to provide the detailed configurations (the number of channels of each layer) of the pruned models, which will strengthen the paper.
>
> We have provided the detailed configuration of all the pruned models of Tab.1 in Appendix L.
>
>
> [1] Alvarez, Jose M., and Mathieu Salzmann. "Learning the number of neurons in deep networks." Advances in Neural Information Processing Systems. 2016.
> [2] Molchanov, Pavlo, et al. "Importance estimation for neural network pruning." Proceedings of the IEEE/CVF Conference on Computer Vision and Pattern Recognition. 2019.
> [3] Yu, Jiahui, and Thomas Huang. "Autoslim: Towards one-shot architecture search for channel numbers." Neural Information Processing Systems Workshop (2019).
> [4] Li, Bailin, et al. "Eagleeye: Fast sub-net evaluation for efficient neural network pruning." European Conference on Computer Vision. Springer, Cham, 2020.

---

> > ### Comment · Reviewer_ywGD · 2021-11-20
> > **Feedback on rebuttal**
> >
> > I have read all the comments and thank the authors for their replies. The authors have addressed my concerns. I tend to raise my score.

---

> ### Comment · Area_Chair_KFGR · 2021-11-20
> **Rebuttal deadline approaching soon**
>
> Dear Reviewer ywGD,
>
> Could you please go over the response from the authors and provide feedback? The rebuttal deadline is approaching soon (November 22nd) and the authors cannot edit the paper after the deadline.
>
> Thanks,
> Area Chair.

---

### Official Review · Reviewer_DgPf · 2021-11-02

**Correctness:** 4
**Technical Novelty And Significance:** 4
**Empirical Novelty And Significance:** 4
**Recommendation:** 6
**Confidence:** 5

**Main Review:**


==+== B. Strengths

The evaluation is strong and the analysis is thorough. The Pareto curves show the effectiveness of the algorithm.
The writing is clean.

==+== C. Weaknesses

In my opinion, the novelty is not very exciting.
There are a lot of design choices (e.g., computing important score) and the alternatives are not discussed.

Questions:

Can you make a comparison between dynamic channel pruning (e.g., https://arxiv.org/pdf/1810.05331.pdf)? The term `dynamic grouping’ is confusing in your paper, you should clarify that it happens during training but not evaluation. Also, my intuition is that using dynamic pruning during evaluation can push the Pareto curves further (fig.3); so it should be an important baseline.

There is also a lot of design options in each part of your design. For example: 1) Computing the importance score you can use the method here (https://arxiv.org/abs/1810.02340) by computing the gradient of each neuron. 2) Regarding the latency prediction, you mentioned that latency is not linear to the number of FLOP; however, for a layer-wise model architecture (most of the models in your evaluation are layer-wise), this is not always true. I am wondering in which scenario latency and FLOPs are not linearly related.

In Table 3, are you sure the baselines (~8000h) are fair compared to yours (30m)? As far as I can tell, you did not include the extra GPU training time. The number reported is confusing, can you possibly show a time breakdown of your compilation process?

Minor:
Your title is so vague and it looks like a survey paper. You can consider changing the name of your paper.

**Summary Of The Paper:**

==+== A. Paper summary


This paper proposes a method to do channel pruning. By building up latency and important score for each group (or channel), the paper use an ILP solver to select the channels that can achieve the best trade-off between latency v.s. Accuracy. The evaluation shows that the method achieves better speedup compared to traditional channel pruning with almost no accuracy loss.


**Summary Of The Review:**

The quality of this paper is good. The experiments are thorough and solid. However, I think the novelty of this paper is limited.

---

> ### Author Response · Authors · 2021-11-18
> **Rebuttal by paper723 Authors**
>
> Dear Reviewer DgPf,
>
> Thank you for the valuable comments and suggestions. Below we provide clarifications to your comments and we have also integrated them in the main paper and appendixes (see paper update).
>
> C1. Can you make a comparison between dynamic channel pruning (e.g., https://arxiv.org/pdf/1810.05331.pdf)? The term `dynamic grouping’ is confusing in your paper, you should clarify that it happens during training but not evaluation.
>
> For comparison to this baseline, in Tab. 1 in the main paper, we compare our VGG16 results with those in the reference provided. As shown, our approach yielded higher accuracy with fewer FLOPs. Precisely, for a 30% pruning ratio, we obtain 72.3% top-1 accuracy with 4.6G FLOPS vs 71.2% with 5.1 G FLOPS. For a 20% pruning ratio, we slightly outperform the accuracy in the reference with a slightly lower number of FLOPS. As the reference did not release any model, we are not able to compare the actual latency on the same GPU.
>
> We agree with the reviewer that dynamic channel pruning might bring further performance boost and we take the dynamic HALP as part of our future work.
>
> C2. 1) Computing the importance score you can use the method here (https://arxiv.org/abs/1810.02340) by computing the gradient of each neuron. 2) Regarding the latency prediction, you mentioned that latency is not linear to the number of FLOPs; however, for a layer-wise model architecture, this is not always true. I am wondering in which scenario latency and FLOPs are not linearly related
>
>
> 1) We formulate neural network pruning as a resource allocation problem where we make the formulation generic to fit a cost constraint and a neuron important measure. As pointed out by the reviewer, there are multiple ways to calculate the importance score of the neurons. In our case, without loss of generality, we used the first-order Taylor expansion, which is calculated from the weights and the corresponding gradients, as it demonstrated great results in [1]. The paper suggested by the reviewer calculates the gradient with respect to the auxiliary variables that function as a mask. In our case, we calculate the gradient with respect to the BN weights. In any case, the key idea behind the two methods is essentially the same.
>
>
>
> In addition, following the reviewer’s suggestion, we instantiate our algorithm using other score functions. In Appendix M (Tab. 11), we show results using L2 norm. As shown, using Taylor expansions outperforms this magnitude-based method. For a similar latency, importance score yields a 1% higher top-1 accuracy (Importance: 76.7% @ 1672fps vs Magnitude: 75.7%  @ 1595fps). Importantly, our proposed HALP algorithm is generic and could be used with other scoring functions.
>
>
> 2) FLOPs have been widely used as a proxy of latency at inference time; however, they are not equivalent. The latency on a GPU usually imposes staircase-shaped patterns for convolution operations with varying channels. In contrast, FLOPs will change linearly. We have clarified this in Appending H. For instance, in Fig.9, we can see that even within one single convolution layer, the actual latency and FLOPs are not linearly correlated.
>
>
>
> C3. In Table 3, are you sure the baselines (~8000h) are fair compared to yours (30m)? The number reported is confusing, can you possibly show a time breakdown of your compilation process?
>
> In Table 3 in the main paper, we provide the time different methods spend on the sub-network selection. After the sub-network selection finishes, all the methods need to spend additional time finetuning the network, which is not taken into consideration.
>
>
> In the paper, we used sub-network selection timing from EagleEye[3]. We actually found out that the numbers reported in [3] refer to the time for choosing a subnet from 1000 candidates while the original paper does not need to generate 1000 candidates.
>
> Following the reviewer’s suggestion, and for a fair comparison, we have updated Tab. 3 including the additional time required during the forward/backward pass to compute the importance score.  We also re-approximate the time cost for all the other methods as they were running on the same device. A breakdown of the execution process for all the methods is added in Appendix O. As shown in the table, with these modifications, our approach is around 4.3x faster than EagleEye[3], the previous state-of-the-art method.
>
> [1] Molchanov, Pavlo, et al. "Importance estimation for neural network pruning." CVPR 2019.
> [2] Lee, Namhoon, Thalaiyasingam Ajanthan, and Philip HS Torr. "Snip: Single-shot network pruning based on connection sensitivity." ICLR 2019.
> [3] Li, Bailin, et al. "Eagleeye: Fast sub-net evaluation for efficient neural network pruning." ECCV, 2020.

---

### Official Review · Reviewer_2vJd · 2021-11-02

**Correctness:** 4
**Technical Novelty And Significance:** 3
**Empirical Novelty And Significance:** 3
**Recommendation:** 6
**Confidence:** 4

**Main Review:**

Importantly, the paper works on a very important topic. Model compression is very important when it comes to deployment to ensure QoS of the inference services.

I believe the main benefits of the work are (1) formulating hardware-aware pruning while retaining good performance and (2) doing this efficiently. The formulation of the overall pruning procedure into a Knapsack problem seems to be where the main benefit comes from. Neuron grouping proposed in the paper also seems to be another source of speedup.

Questions I have are:

* While the paper claims that this can be applied to other platforms too. However, this requires re-generating the look-up table for HALP. Could you provide how long this would take for a new platform? Can this be done efficiently?
* How would this perform in relation to quantization? And, would similar mechanism for enforcing hardware-awareness in compression work for quantization?

**Summary Of The Paper:**

This paper is an effort to perform structured pruning with regard to inference latency on hardware. The paper solves this by creating a look-up table of "importance score" and "latency contribution" and formulates this into a Knapsack problem, which can be solved efficiently. Evaluation shows that this method effectively reduces the latency with regard to hardware via pruning.

**Summary Of The Review:**

Hardware-awareness of pruning, while touched upon by some works, still opens up large potential for inference latency reduction. This paper proposes an interesting direction in optimization where each layer is assigned importance score and measured latency of each layer. Then, the overall pruning problem is formulated as a knapsack problem. Overall performance of the pruned network seems reasonable. I liked reading the paper, and I would like the paper to be in the program.

---

> ### Author Response · Authors · 2021-11-18
> **Rebuttal by paper723 Authors**
>
> Dear Reviewer 2vJd,
>
> Thank you for the valuable comments and suggestions. Below are our answers to your comments and details regarding the modifications to the main paper and appendix.
>
>
> C1. While the paper claims that this can be applied to other platforms too. However, this requires re-generating the look-up table for HALP. Could you provide how long this would take for a new platform? Can this be done efficiently?
>
>
> Thank you for pointing this out. As also replied to reviewer dTnS,  we provide a more detailed description of the algorithm to generate the look-up table in
> Appendix N. It takes about 1 hour to generate the table for the MobileNet and approximately 5 hours for the ResNet family. As different networks within the same architecture family share similar layer structures, the LUT can be reused (as in Resnet50 and Resnet101) or mostly reused. Importantly, once the LUT is generated it can be reused for all the possible target latencies for that architecture on that platform.

---

> > ### Comment · Reviewer_2vJd · 2021-11-19
> > **I stand by my score**
> >
> > I am satisfied with the answer. However, I think it would be nice if some of these details such as how long it takes are incorporated somewhere in the paper. Possibly the appendix.
> >
> > As a constructive feedback, I think some research on how this process can be shortened, approximated, reused would make this stronger.

---

> > > ### Author Response · Authors · 2021-11-22
> > > **Additional answers to reviewer's comments**
> > >
> > > Dear Reviewer 2vJd,
> > >
> > > Thank you for your response and positive feedback. The details about the time spent on latency look-up table creation, as well as the creation process, has already added to the revised version of the manuscript as Appendix N. Please, refer to the newer version of the submission.
> > >
> > > We also address your other question regarding the compression for quantization in Appendix Q.
> > >
> > > C2. How would this perform in relation to quantization? And, would similar mechanism for enforcing hardware-awareness in compression work for quantization?
> > >
> > > In Appendix P in the revised version of the manuscript, we provide ResNet50 pruning results for INT8 quantization. Precisely, we generated the lookup table using NVIDIA Xavier platform to measure the actual latency for INT8 networks. For comparison, we create both a INT8 and a FP32 latency look-up table. Then, we quantize the output of our algorithm into INT8 for inference. As shown in Fig.14, even with a FP32 latency look-up table, HALP outperforms the state-of-the-art method EagleEye [1]. When using the INT8 look-up table, our method yields even better results. We get a 1.32x speedup while maintaining the original Top1 accuracy. Moreover, compared to EagleEye [1], the current state of the art, our approach yields up to 1.26x relative speedup and 0.15% higher accuracy.
> > >
> > > [1] Li, Bailin, et al. "Eagleeye: Fast sub-net evaluation for efficient neural network pruning." European Conference on Computer Vision. Springer, Cham, 2020.

---

### Official Review · Reviewer_dTnS · 2021-11-02

**Correctness:** 3
**Technical Novelty And Significance:** 2
**Empirical Novelty And Significance:** 2
**Recommendation:** 5
**Confidence:** 3

**Main Review:**

Strengths

- This paper is well-written, well-motivated, and clear presentation.
- The proposed method improves network throughput efficiency with competitive accuracy on classification and detection tasks, outperforming prior pruning approaches.
- Formulating the hardware-aware structural pruning as a knapsack problem is new.


Weaknesses

- The reviewer's main concern is that this work is one of the lookup-table(LUT)-dependent methods that is not new and has critical limitations.
    - After the emerging MNasNet[1], there are already numerous LUT-based methods improving the efficiency of neural networks under latency-constrained of each target hardwares.
    - For the case that the correlation between the predicted latencies by LUT and the real latencies is low, the performance of LUT-dependent methods becomes poor. That means the performance of LUT is critical rather than the proposed method. The reviewer guesses that this method also would not guarantee high performance on the target tasks when the performance of LUT is poor.
    - For considering many hardwares, building LUT process for all hardwares repetitively is not negligible and becomes a heavy burden. We should build LUT for each hardware and we should compile neural networks dependent on device platform types (sometimes it needs domain knowledge of experts).
    - In addition, the information about LUT such as building process, building time, and correlation between the estimated and the real ones looks omitted in the paper.
- Generality Issue on multiple hardwares. While this method is 'hardware-aware', the target devices used in the experiments are just two (GPU and Jetson). Recent hardware-aware methods for the efficiency of neural networks under latency constraints have been handled many hardware platforms and device entities. For example, OFA[2] considered 'GPU/CPU/Mobile Phone/Edge GPU/FPGA, BRP-NAS[3] considered 'GPU/CPU/Mobile Phone/Edge GPU/Edge TPU, and HELP[4] considered 'GPU/CPU/Mobile Phone/Edge GPU/FPGA/AISC/Raspberry Pi.

[1] Mnasnet: Platform-aware neural architecture search for mobile, CVPR2019.

[2] Once-for-all: Train one network and specialize it for efficient deployment, ICLR2020.

[3] Brp-nas: Prediction-based nas using gcns, NeurIPS2020.

[4] HELP: Hardware-Adaptive Efficient Latency Predictor for NAS via Meta-Learning, NeurIPS 2021.

**Summary Of The Paper:**

This work tackles a latency-constrained structural pruning method by formulating the global resource allocation optimization problem and addressing them via an augmented knapsack solver. During the pruning process, to estimate latency on a target device and accuracy drop, this work uses look up table and global saliency score, respectively. The proposed approach is validated on classification and detection tasks with desktop GPU (Titan V) as the target device.

**Summary Of The Review:**

This paper has a clear motivation and introduces a knapsack algorithm for NAS, yet, validates the proposed method on limited devices and has weak novelty since the latency estimation is dependent on LUT.

---

> ### Author Response · Authors · 2021-11-18
> **Rebuttal by paper723 Authors**
>
> Dear Reviewer dTnS,
>
> Thank you for the valuable comments and suggestions. Below are our answers to your comments and details regarding the modifications to the main paper and appendix.
>
> We agree with the reviewer that any LUT-based pruning methods (including ours) heavily rely on the generation of the LUT. In Appendix N, Fig. 13, we provide a correlation analysis between the LUT predicted latencies and the actual latencies of the models. As we can see, there is a strong linear correlation between them.
>
> We also provide a more detailed description of the algorithm to generate the look-up table in the same appendix. In our case, it takes about 1 hour to generate the table for the MobileNet and approximately 5 hours for the ResNet family. As different networks within the same architecture family share similar layer structures, the LUT can be greatly reused (as in Resnet50 and Resnet101).
>
> As pointed out by the reviewer, different hardware platforms require regenerating the LUT. Nevertheless, once the table has been generated it can be reused for all the experiments on that platform and architecture.

---

> > ### Author Response · Authors · 2021-11-22
> > **Additional comments to reviewer's comments**
> >
> > Dear Reviewer dTnS,
> >
> > As suggested, we added additional pruning experiments on two other hardware platforms: Intel CPU Xeon E5 and NVIDIA Xavier. Detailed results for these two new hardware platforms can be found in the Appendix P of the revised version of the submission.
> >
> > As we show in Fig.14 in the Appendix P, our proposed HALP algorithm consistently yields better performance compared to the state-of-the-art method EagleEye [1]. By leveraging latency characteristics on different hardware, HALP makes a better trade-off between the accuracy and efficiency. Specifically, compared to the original model on the Intel CPU, HALP yields 1.27X speedup with a 0.15% increment in top-1 accuracy; On the NVIDIA Xavier platform, HALP yields a 1.23X speedup with a 0.28% increment in top-1 accuracy. Compared to EagleEye, HALP achieves up to 1.45X speedup with a similar accuracy.  From these results, we can conclude that our method generalizes well to different hardware.
> >
> > [1] Li, Bailin, et al. "Eagleeye: Fast sub-net evaluation for efficient neural network pruning." European Conference on Computer Vision. Springer, Cham, 2020.

---

### Author Response · Authors · 2021-11-22
**Author response to All reviewers**

Dear reviewers and ACs,

We sincerely thank all the reviewers for their valuable comments and suggestions to help improving the paper. We have integrated all the comments into our main paper and additional appendixes. The main changes of the paper are summarized as below:

- Results of pruning a MobileNet-V2 is added in Tab.1 of main paper and Appendix K.
- Algo.1 of main paper is updated to be more rigorous. A detailed explanation of Algo.1 is added in Appendix I.
- Tab.3 of main paper is updated for a fair comparison of the extra computation required by different pruning methods. A detailed time breakdown is added in Appendix O.
- We add the ablation study of pruning step k in Appendix J.
- The detailed structure of the pruned models is shown in Appendix L.
- A plot showing the non-linearity between latency and FLOPs is added as Fig.9 in Appendix H.
- We show the ablation study of using different importance calculation in Appendix M.
- We add the details of the latency look-up table creation process, as well as the look-up table building time in Appendix N. We also show the correlation between the predicted latency from look-up table and the real latency in the same appendix. - Additional results targeting on other two hardware, Intel CPU Xeon E5 and NVIDIA Xavier, are added in Appendix P.
- We provide pruning results for INT8 quantization in Appendix Q.

---

### Decision · Program_Chairs · 2022-01-20

**Decision:**

Reject

**Comment:**

This paper proposes a hardware-aware pruning method which structurally prunes the given deep neural networks to retain their accuracy while satisfying the latency constraints. Specifically, the authors formulate the latency-constrained pruning problem as a combinatorial optimization problem to find the optimal combination of neurons to maximize the sum of the importance scores, and propose an augmented knapsack solver to solve it, as well as a neuron grouping technique to speed up the training. The proposed method is validated for its classification tasks on two devices, namely Titan V and Jetson TX2, and for object detection performance on Titan V, and is shown to achieve superior accuracy/latency tradeoff compared to existing pruning methods, including latency-aware ones.

The paper received split reviews initially, and the following is the summary of the pros and cons mentioned by the reviewers.

Pros
- The proposed formulation of the latency-constrained pruning problem as a constrained knapsack problem is novel.
- The method achieves competitive performance against existing latency-constrained pruning methods.
- The paper is written well, with clear motivation and descriptions of the proposed method.

Cons
-  The idea is not very exciting since posing pruning as a combinatorial optimization problem, or a knapsack problem is not new, and the proposed method only adds in additional latency constraints.
- The title “hardware-aware” is vague and misleading since what the authors do are latency-constrained pruning.
- The experimental validation is only done on two devices, which makes the method less convincing as a “hardware-aware” method and how it generalizes to other devices (e.g. CPU, FPGA)
- Use of lookup tables to obtain the latency constraints is not novel, has a limited scalability, and is inefficient.
- Missing discussion of design choices.

During the discussion period, the authors cleared away some of the concerns, which resulted in two of the reviewers increasing their scores. However, one reviewer maintained the negative rating of 5, and the positive reviewers were still concerned with limited novelty.

I believe that this is a good paper that proposes a neat solution for latency pruning, which may have some practical impact. However, the novelty of the idea is limited, as pointed out by the reviewers. The use of lookup tables also does not seem to be an efficient solution for adapting to edge devices for which the collection of latency measurements could be slow. The experimental validation on only two devices of the same type (GPU) also seems insufficient, as how the method generalizes to diverse devices is uncertain. It would be worthwhile to consider using a latency predictor (e.g. BRP-NAS [Dudziak et al. 20]), and perform experimental validation on diverse hardware platforms (e.g. CPU and FPGA). Comparing against recently proposed hardware-aware NAS methods could be also interesting, as there has been a rapid progress on the topic recently.

Thus, despite the overall practicality and the quality of the paper, the paper may benefit from another round of revision, since both the method and the experimental validation part could be improved.

[Dudziak et al. 20] BRP-NAS: Prediction-based NAS using GCNs, NeurIPS 2020